# Genome-wide signatures of adaptation to extreme environments in red algae

Chung Hyun Cho [1], Seung In Park[1], Tzu-Yen Huang[1], Yongsung Lee[1], Claudia Ciniglia[2], Hari Chandana Yadavalli[3], Seong Wook Yang [3], Debashish Bhattacharya[4] & Hwan Su Yoon [1] ✉

The high temperature, acidity, and heavy metal-rich environments associated with hot springs have a major impact on biological processes in resident cells. One group of photosynthetic eukaryotes, the Cyanidiophyceae (Rhodophyta), has successfully thrived in hot springs and associated sites worldwide for more than 1 billion years. Here, we analyze chromosome-level assemblies from three representative Cyanidiophyceae species to study environmental adaptation at the genomic level. We find that subtelomeric gene duplication of functional genes and loss of canonical eukaryotic traits played a major role in environmental adaptation, in addition to horizontal gene transfer events. Shared responses to environmental stress exist in Cyanidiales and Galdieriales, however, most of the adaptive genes (e.g., for arsenic detoxification) evolved independently in these lineages. Our results underline the power of local selection to shape eukaryotic genomes that may face vastly different stresses in adjacent, extreme microhabitats.

Over long evolutionary history, species have adapted to a wide range of extreme conditions, and these environments continue to host a biodiverse microbial community[1,2]. Notably, organisms inhabiting extreme environments (i.e., so-called, extremophiles) face significant physical (e.g., atmospheric pressure, solar radiation, and temperature) and geochemical stresses (e.g., desiccation, oxygen levels, pH, salinity, and redox potential) that place strict limits on metabolic functions[3]. Given these strong selective forces, species have adopted three strategies to overcome external challenges from the environment: 1) establishing a novel, beneficial system (e.g., through horizontal gene transfer [HGT]), 2) discarding ancestral traits to avoid energy waste (e.g., genome reduction), and 3) modifying the ancestral system to be more robust (e.g., altering the thermostability of proteins)[3–6]. When these factors are considered, genomic data from extremophiles have the potential to elucidate evolutionary transitions that result from temperature, pH, salinity, and other stresses when compared to mesophilic lineages[7].

The red algal class, Cyanidiophyceae, was once (mistakenly) described as the most primitive eukaryotic microbes with "pro-eukaryotic features", which refers to early eukaryotic traits based on physiological and morphological characteristics[8]. We now know with some certainty that extremophily is a derived trait in the Cyanidiophyceae, which shares a common ancestry with mesophilic Archaeplastida[6,9]. These unicellular red algae thrive in a wide range of high-temperature (>50 °C), acidic (~pH 1), and heavy metal-rich environments that are lethal to most eukaryotes, and Cyanidiophyceae comprise nearly all of the eukaryotic biomass present in these areas[10–13]. Exotic prokaryotic genes have recently been discovered in the nuclear genomes of these algae, allowing them to inhabit a variety of extreme habitats[9,14,15]. For example, analysis of the *Galdieria* 074W genome identified an ATPase (adenosine triphosphatases) gene derived from Archaea that underwent subsequent duplication events[14]. Based on these studies, Cyanidiophyceae are excellent eukaryotic models (particularly, *Cyanidioschyzon*, for which genetic tools exist) for studying the relationship between environmental adaptation and genome evolution[16,17].

Several lines of evidence (e.g., phylogeny, morphological traits, ecological habitats, and energy production systems) suggest that the

---

[1]Department of Biological Sciences, Sungkyunkwan University, Suwon 16419, Korea. [2]Department of Environmental, Biological and Pharmaceutical Science and Technologies, University of Campania Luigi Vanvitelli, Caserta, Italy. [3]Department of Systems Biology, Institute of Life Science and Biotechnology, Yonsei University, Seoul, Korea. [4]Department of Biochemistry and Microbiology, Rutgers University, New Brunswick, NJ, USA. ✉e-mail: hsyoon2011@skku.edu

Cyanidiophyceae is divided into two major orders, the Cyanidiales and Galdieriales (previously Cyanidiaceae and Galdieriaceae)[18,19]. Draft genome assemblies are currently available for 14 cyanidiophyceans: one strain of *Cyanidioschyzon merolae*, two strains of *Cyanidiococcus yangmingshanensis*, nine strains of *Galdieria sulphuraria*, and two strains of *Galdieria phlegrea*[9,14,15,20,21]. Genomic studies of Cyanidiophyceae have been largely limited to *Galdieria* (11 out of 14) and much less is known about the Cyanidiales. None of the available Galdieriales genomes are at the chromosome-level. Consequently, technical issues such as inaccurate or incomplete gene models, taxonomic misidentification (e.g., *Cyanidioschyzon* sp. for *C. yangmingshanensis* Soos strain), and DNA contamination have been reported (Supplementary Note 1; Supplementary Figs. 1, 2). The development of long read-based assembly and diverse scaffolding methods have enabled the generation of telomere-to-telomere (T2T) genomes[22]. T2T or near-chromosome-level assemblies provide many insights at the genome-scale, ranging from structural evolution (e.g., genome duplication) to population history (e.g., introgression), and allow the design of epigenomic studies or quantitative trait locus (QTL) analysis[23–26]. Despite dozens of draft genomes being available for Cyanidiophyceae, there remain major limitations to their use, reflecting biased taxon sampling, an unclear taxonomy, and errors associated with genome assembly and gene prediction, which have hindered understanding of this fascinating lineage.

Given these limitations, we generated three chromosome-level genome assemblies, two from Cyanidiales and one from Galdieriales species. Our analyses demonstrate that genes related to specific environmental stresses (e.g., heavy metal detoxification) were acquired through HGT events and independent subtelomeric gene duplication (STGD) enhanced cell resilience in each lineage. We present data that elucidate Cyanidiophyceae genome evolution and shed light on lineage-specific genome changes that demonstrate selection at the microhabitat scale.

## Results and discussion

### Genomes of Galdieriales and Cyanidiales

To investigate genome evolution in Cyanidiophyceae, we generated genome and transcriptome data from two Cyanidiales species, *Cyanidium caldarium* 063 E5 (CDCA; hereafter, *Cyanidium*) and *Cyanidiococcus yangmingshanensis* 8.1.23 F7 (CCYA; hereafter, *Cyanidiococcus*), and one Galdieriales species, *Galdieria sulphuraria* 108.79 E11 (GASU; hereafter, *Galdieria*) (Supplementary Dataset 1). Telomeres were identified in these three genomes, and we discovered that these regions are highly diverse compared to the probable ancestral telomeric repeats (Supplementary Note 2; Supplementary Figs. 3, 4; Supplementary Dataset 2)[27]. We reconstructed the sequence of all, or nearly all of the 20 T2T chromosomes from *Cyanidium* and *Cyanidiococcus*, which have haploid genome sizes of 12.0 Mbp and 8.8 Mbp, respectively (Table 1; Supplementary Dataset 2). Although we did not generate T2T chromosome data from *Galdieria*, the genome formed a 14.5 Mbp assembly with a pseudochromosome-level of 76 scaffolds (58 T2T scaffolds, 16 single-end telomere scaffolds, two scaffolds without telomeres, see Supplementary Note 3; Supplementary Fig. 5; Supplementary Dataset 3). Based on telomeric repeat identification, *Galdieria* appears to have more chromosomes (at least 66) than the 57 chromosomes reported in a pulsed-field gel electrophoresis (PFGE) study[28]. Gene prediction using ab initio modeling with manual curation identified 4870 protein-coding genes (CDSs) in *Cyanidium* and 4832 CDSs in *Cyanidiococcus*, both being spliceosomal intron-poor (<50 introns), whereas the *Galdieria* genome contained 7020 CDSs with an intron-rich gene structure (>10 K introns). Using protein-coding sequences, the BUSCO (i.e., conserved eukaryotic gene inventory) result for *Cyanidium* was 94.7% (C: 87.8%, F: 6.9%), for *Cyanidiococcus* was 96.7% (C: 92.4%, F: 4.3%), and for *Galdieria* was 95.7% (C: 94.0%, F: 1.7%). These data demonstrate the complete nature of these genomes when compared to a BUSCO score of 96.7% (C: 93.4%, F: 3.3%) for the T2T *Cyanidioschyzon* 10D genome.

### Differential evolution of chromosomes in Galdieriales and Cyanidiales

Based on properties such as gene structure (e.g., number of introns and genes) and chromosomal features (e.g., chromosome numbers), Cyanidiales and Galdieriales genomes show high divergence. For example, the two Galdieriales, including *G. sulphuraria* 108.79 E11 and the publicly available *G. sulphuraria* MtSh, contain >3-fold more chromosomes (at least 66 based on genome comparisons and the number of telomere-containing scaffolds, see Supplementary Fig. 5; Supplementary Dataset 3) than the three Cyanidiales, which all contain 20 chromosomes. Average chromosome sizes range from 439.4–827.3 kbp in the three Cyanidiales species, whereas they were about 3-fold smaller (190.9 kbp) for *G. sulphuraria* 108.79 E11. To determine if *Galdieria* genomes show structural conservation with those of Cyanidiales, we compared chromosomal gene synteny. Compared to the three Cyanidiales species, 5–11 out of 65 (excluding 11 incomplete chromosomal scaffolds) gene synteny blocks of *Galdieria* chromosomes partially matched to the Cyanidiales chromosomes, showing that gene order is not strongly conserved between Cyanidiales and Galdieriales (Fig. 1a; Supplementary Fig. 6).

To analyze the Cyanidiales genomes, gene synteny of the two new genomes was compared to the reference genome of *C. merolae* 10D[29]. The three Cyanidiales species share a large number of syntenic blocks (86.0% shared, collinear genes) with dozens of chromosomal recombination events, in particular, between *Cyanidium* and *Cyanidioschyzon* (Fig. 1A). Nine chromosomes were fully conserved between *Cyanidioschyzon* and *Cyanidiococcus*, whereas the remaining 11 chromosomes showed chromosomal divisions, fusions, inversions, and relocations. There was a small difference in gene content between *Cyanidiococcus* and *Cyanidioschyzon* (CCYA: 4832 CDSs; CZME: 4803 CDSs), even though the *Cyanidioschyzon* genome is 1.38x larger than in *Cyanidiococcus* (CCYA: 12.0 Mbp; CZME: 16.5 Mbp) (Table 1). *Cyanidioschyzon* has fewer introns (CCYA: 36 introns, CZME: 27 introns), therefore the genome size difference cannot be explained by intron insertion in *Cyanidioschyzon* (Table 1). Using a statistical approach (Student's $t$-test: $p$-value <0.05), we discovered that the average intergenic region of the three Cyanidiales species (CCYA: 929.7 bp, CZME: 1889.8 bp, CDCA: 319.7 bp) is significantly different among them. The size of intergenic regions between sister species increased due to repeat expansion in *Cyanidioschyzon* (CZME: 2.09 Mbp [12.7% of the genome]; CCYA: 71.5 kbp [0.59% of the genome]) (Supplementary Figs. 7, 8b; Supplementary Dataset 4). The chromosomes of two *Cyanidiococcus* strains are highly conserved, with only a few exceptions including a single inversion in the largest chromosome (CCYA01 chromosome from the genome of the 8.1.23 F7 strain) (Supplementary Fig. 1). Similarly, a few chromosomal recombination events exist among *Galdieria sulphuraria* strains (Supplementary Fig. 5). However, chromosomes in the two strains of *Galdieria* are more diverged (sharing 71.7% gene collinearity) than among the three Cyanidiales genera (86.0%), due to mismatched scaffolds (MtSh_40, 42, and 77) in the MtSh genome.

### Highly conserved subtelomeric regions in Cyanidiophyceae chromosomes

Telomere-containing scaffolds were compared to determine if there were any conserved regions between chromosomes in each species (Supplementary Fig. 8). We found 20–30 kbp regions near telomeric repeats, known as telomere-proximal subtelomeric regions (hereafter, subtelomeres), that are conserved in intraspecies chromosome comparisons with minor variation in gene insertions or deletions (two shaded regions at the end of chromosomes in Supplementary Fig. 8). Even though the structure of subtelomeric regions was different

**Table 1 | Summary of genome traits from representative Cyanidiophyceae**

| Taxonomic group | Cyanidiophyceae (Cyanidiophytina) | | | | | |
|---|---|---|---|---|---|---|
| | Cyanidiales | | | Galdieriales | | |
| Species | CZME 10D | CCYA 8.1.23 F7 | CDCA 063 E5 | GASU 074W | GASU 108.79 E11 | GAPH Soos |
| Genome size (Mbp) | 16.5 | 12.0 | 8.79 | 13.7 | 14.5 | 14.9 |
| GC (%) | 55.0 | 54.6 | 65.7 | 37.7 | 40.2 | 37.5 |
| # scaffolds | 20 (Chr.) | 20 (Chr.) | 20 (Chr.) | 433 | 76 | 108 |
| N50 | – | – | – | 172.3 kbp | 191.6 kbp | 202.1 kbp |
| # telomere-containing scaffolds | 20 pairs | 19 pairs + 1 SE | 20 pairs | 24 SEs | 58 pairs + 16 SEs | 8 pairs + 38 SEs |
| # proteins | 4,803 | 4,832 | 4,870 | 7,174 | 7,021 | 6,125 |
| BUSCO (C + F) | 96.4% | 96.7% | 94.7% | 94.1% | 95.7% | 95.1% |
| # introns | 27 | 36 | 44 | 13,245 | 15,190 | 14,106 |
| Reference | Previous study[29] | This study | This study | Previous study[14] | This study | Previous study[15] |

Only single-end (SE) telomere scaffolds larger than 40 kbp were tabulated. CCYA *Cyanidiococcus yangmingshanensis*, CDCA *Cyanidium caldarium*, CZME *Cyanidioschyzon merolae*, GAPH *Galdieria phlegrea*, GASU *Galdieria sulphuraria*.

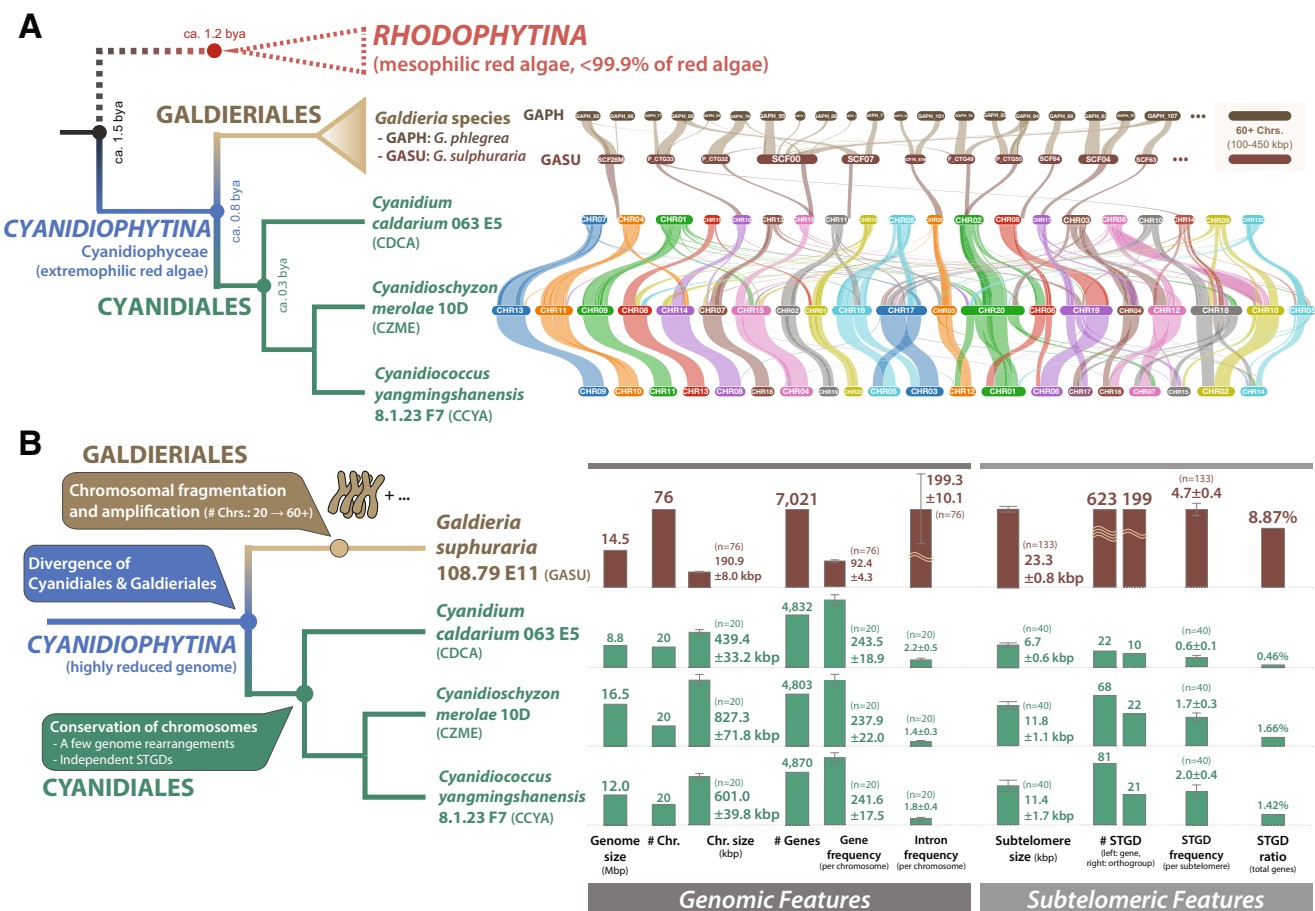

**Fig. 1 | Schematic image showing chromosome evolution and structural variation in Cyanidiophyceae. A** Comparison of gene synteny among chromosomes of cyanidiophycean species. Only chromosomes with conserved gene synteny with Cyanidiales are shown for Galdieriales. **B** Genomic and subtelomeric characteristics of Cyanidiophyceae are compared and summarized using bar plots with standard error bars. Subtelomeric gene duplication (STGD) events are more prevalent in Galdieriales than in Cyanidiales. Bar plots with standard error bars were used to depict the mean values of each feature.

between Cyanidiales and Galdieriales, we identified some common features (see 'Subtelomeric Features' in Fig. 1B; more details in Supplementary Datasets 5, 6). A total of 133 subtelomeric regions were identified from 76 scaffolds of *Galdieria* 108.79 E11 and 40 subtelomeric regions from each Cyanidiales genome. Not only was the number of subtelomeres higher in *Galdieria*, but their cumulative size was 2–4 times larger than in other Cyanidiales species (Fig. 1B). Accordingly, genes located in the subtelomeric regions showed significant differences: 22–81 genes in the three Cyanidiales species, and 623 genes in *Galdieria* 108.79 E11. Expansion of the subtelomeric region in terms of size and the number of encoded genes provides evidence of gene duplications that comprise a larger proportion of the

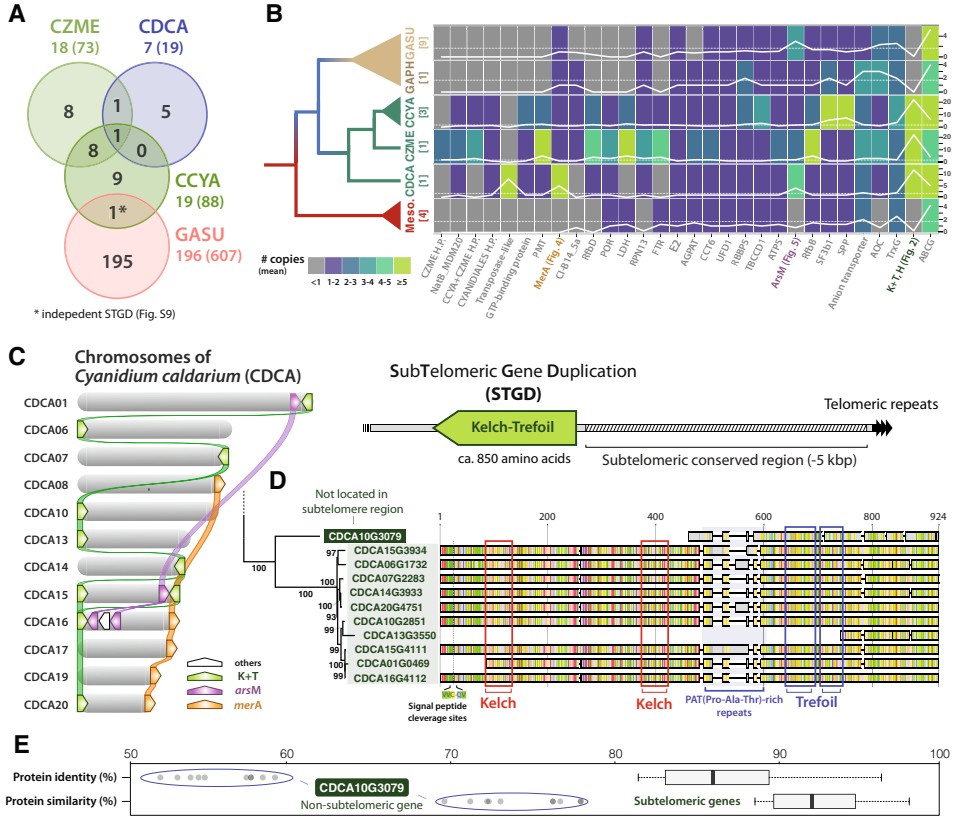

**Fig. 2 | Subtelomeric gene duplications (STGDs) in Cyanidiales. A** The STGD orthogroup Venn diagram of four cyanidiophycean species. The number of orthogroups was indicated, along with number of genes in the STGD orthogroup denoted in brackets. **B** Copy numbers of Cyanidiales subtelomeric duplicated gene family in all red algal species. All duplicated genes were counted including non-STGD genes. The number next to taxa refers to the number of species or strains. Meso.: mesophilic red algae. **C** Characterization of STGDs and featured STGDs in the chromosomes of *Cyanidium caldarium* 063 E5; *ars*M (arsenic methyltransferase), *mer*A (mercuric reductase), and Kelch & Trefoil (K+T) domain containing protein-coding genes. **D** Protein alignment and phylogenetic relationship of K+T domain containing genes in *Cyanidium*. The alignment included annotations for identified domains (e.g., kelch, trefoil) and specialized peptides (e.g., signal peptide). **E** Protein similarities and identities in *Cyanidium* K+T containing proteins were visualized using a bar plot. Excluding the comparison with a non-subtelomeric gene ('CDCA10G3079'), all the genes are highly conserved with 80% of the above protein identities. Box and whisker plots, which highlight the 25th to 75th percentiles of the data and draw a thick line through the mean value, were used to represent the data ranges (*n* = 55).

gene inventory in these regions in *Galdieria* (8.87%) than in Cyanidiales species (0.46–1.66%) (Fig. 1B). This result supports the idea that chromosome fragmentation-mediated subtelomeric gene duplication resulted in a larger number of duplicated genes in Galdieriales.

After the identification of the subtelomeric regions in each species, interspecies comparisons revealed that subtelomeric regions evolved in a species-specific manner. Except for phylogenetically closely related genera such as *Cyanidioschyzon* and *Cyanidiococcus*, subtelomeres could not be aligned among cyanidiophycean species (i.e., *Cyanidium* vs. *Galdieria*), implying a lack of sequence homology. In addition to subtelomeric duplications, non-subtelomeric duplication areas (yellow-colored regions in Supplementary Fig. 8d) were found only in Galdieriales chromosomes. Thus, duplication of syntenic regions in subtelomeric regions may have increased the number of genes in Galdieriales.

### Investigation of subtelomeric gene duplications (STGDs) in Cyanidiophyceae

Given the finding of conserved subtelomeric regions among chromosomes within a species, it is apparent that genes in these regions spread to the subtelomeric regions of other chromosomes. This feature was observed in many chromosomes, and we refer to these as STGDs (turquoise and red block arrow in Supplementary Fig. 8). STGDs in Cyanidiales and Galdieriales are clearly different, both in number, Galdieriales have more STGDs than Cyanidiales (Fig. 2A;

Galdieriales: 607 genes, Cyanidiales: 19–88 genes) and the fractional proportion of the total gene inventory (Galdieriales, 8.87%; Cyanidiales, 0.46-1.66%) (Fig. 1B; see details in Supplementary Datasets 5-7). We identified 228 orthogroups from all Cyanidiales and Galdieriales STGD families. However, most of the STGDs are not shared between these two lineages (Fig. 2A; Supplementary Dataset 7). GTP-binding protein STGDs were found in both *Galdieria* and *Cyanidiococcus*, although they appear to have been duplicated independently in each lineage (Supplementary Fig. 9) after divergence. Except for a single orthogroup that contains genes for the kelch, trefoil, and hedgehog domains (see details below), none of the commonly shared subtelomeric gene duplication events are present in the three T2T Cyanidiales genomes. This result indicates either subtelomeric duplicated genes in some chromosomes are distinct from those in other chromosomes in the last common ancestor of Cyanidiales, followed by differential inheritance of subtelomeric duplicated genes into the two Cyanidiales lineages, or that STGDs occurred post-divergence of this order. The STGD ratio was calculated to determine its impact on recent gene duplication events (Supplementary Fig. 10). STGDs accounted for 28.9–31.9% of recent gene duplications in both Cyanidiales and Galdieriales, and Fisher's exact test (*p*-value 0.05) supported the correlation between gene duplication and subtelomeric region. As a result, recent gene duplications of Cyanidiophyceae species have been significantly influenced by STGD events.

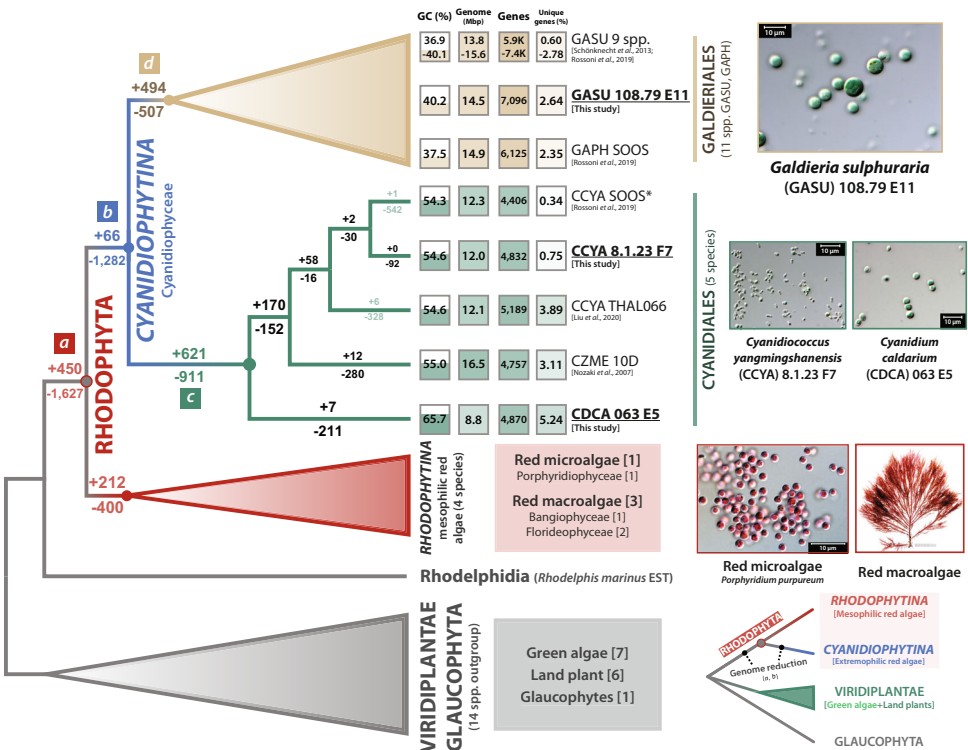

**Fig. 3 | Overview of gene family gains and losses during Cyanidiophyceae evolution.** A parsimony approach was used to estimate the gain and loss patterns of orthologous gene families, based on the 36 representative taxa used in this study. Two major genome reduction events (>1 K gene families were lost) occurred in the cyanidiophycean lineage. Most gene gains (e.g., HGTs) and losses occurred independently during the diversification of Cyanidiales and Galdieriales. a: Divergence of red algae, b: Divergence of Cyanidiophyceae, c: Divergence of Cyanidiales, d: Divergence of Galdieriales.

At lower taxonomic levels, there are more shared subtelomeric genes. The average size of subtelomeric regions in Cyanidiales genomes was around 6.7–11.8 kbp (Fig. 1B), but some subtelomeric regions were found to be as large as 44 kbp (e.g., *Cyanidiococcus* CCYA08 chromosome). Except for unidentified proteins with ambiguous functions, 28 different types of subtelomeric duplicated genes were discovered in Cyanidiales (Fig. 2B, C; Supplementary Fig. 11; Supplementary Datasets 5, 7), but these were not identical to subtelomeric duplicated genes in Galdieriales. The most prevalent STGDs in Cyanidiales were a kelch domain (K; identified as galactose oxidase) fused with a trefoil domain (T; identified as trefoil factor in *Cyanidioschyzon*) or a hint domain (H; identified as hedgehog proteins in *Cyanidioschyzon*) connected by a threonine (Thr)/proline (Pro)/alanine (Ala)-rich conserved peptide (36.1% of Thr, 18.6% of Pro, 12.6% of Ala). Around 12–24 copies of K, T, H domain-coding genes are present in the subtelomeric regions of Cyanidiales species (Fig. 2D). Each Cyanidiales species has different combinations of K, T, and H domains (e.g., K, T, H, K+T, K+H) and uniquely duplicated domains in the subtelomeric regions (Fig. 2C). In the *Cyanidium* genome, most of the K, T, and H genes show low variation (>80% of protein identity), but a trefoil domain-only containing gene (e.g., CDCA10G3079) that was not located in the subtelomeric region, has lower protein identity (50–60%) with other homologs (Fig. 2E). Kelch-hint domain fused genes (K+H; identified as hedgehog proteins in *Cyanidioschyzon*) comprise three copies in *Cyanidiococcus* and 11 copies in *Cyanidioschyzon*, whereas K+H fused genes were not identified in *Cyanidium*. Because the function of the kelch domain is highly diverse: i.e., extracellular communication/interaction, cell morphology, gene expression, actin binding, and virus post-infection[30], it is not possible to assign specific functions to kelch domains in Cyanidiales. Another interesting feature in this order is a linker peptide that connects two major domains made up of threonine/proline/alanine-rich repeats (up to 365 amino acids in *Cyanidiococcus*). Furthermore, size variation of the tripeptide repeats (spacer sequences; linker peptides) was observed among subtelomeric duplicated proteins, and these linker peptides may promote divergence of protein functions[31,32]. However, we are currently unable to determine the selective benefits of K, T, H variants derived by STGDs.

*Cyanidioschyzon* and *Cyanidiococcus*, two closely related species, share the majority of STGDs and conservation of duplicated genes, which were not observed in *Cyanidium* chromosomes. For example, synteny blocks encoding five protein-coding genes (PMT, RPN13, iron permease, RfbB, RfbD) within subtelomeric regions were discovered in four chromosomes of *Cyanidioschyzon* (CZME10, 12, 14, 17) and two chromosomes of *Cyanidiococcus* (CCYA08, 15) with some minor variations (Supplementary Fig. 11).

We studied the evolutionary pressure on Cyanidiales-conserved subtelomeric genes, which were suggested to be a target for rapid adaptive evolution[33]. Although the $K_a/K_s$ ratios of a few pairs did not pass Fisher's exact test (7 out of 21 pairs; $p$-value≤0.05), due to the small number of nucleotide substitutions (3–8 changes out of 1,452-1,476 bp) from subtelomeric duplications, a few interspecies gene pairs show more evidence of purifying selection than subtelomeric duplicated gene pairs under relaxed or positive selection ($K_a/K_s$ average of 11 interspecies *mer*A pairs: 0.10, $K_a/K_s$ average of 10 duplicated *mer*A pairs: 5.60; Supplementary Dataset 8), according to the $K_a/K_s$ ratio of Cyanidiales genes. We were also able to observe that H3K27me3 histone modifications were highly enriched in (sub)telomeric regions by reanalyzing ChIP-seq data from a previous study (Supplementary Fig. 12)[33]. H3K27me3 modifications may play a role in regulating gene activation.

By investigating subtelomeric regions of cyanidiophycean genomes, including published data[29], we discovered some essential genes in these regions that are associated with environmental adaptation in extremophiles. Most of the subtelomere-located genes

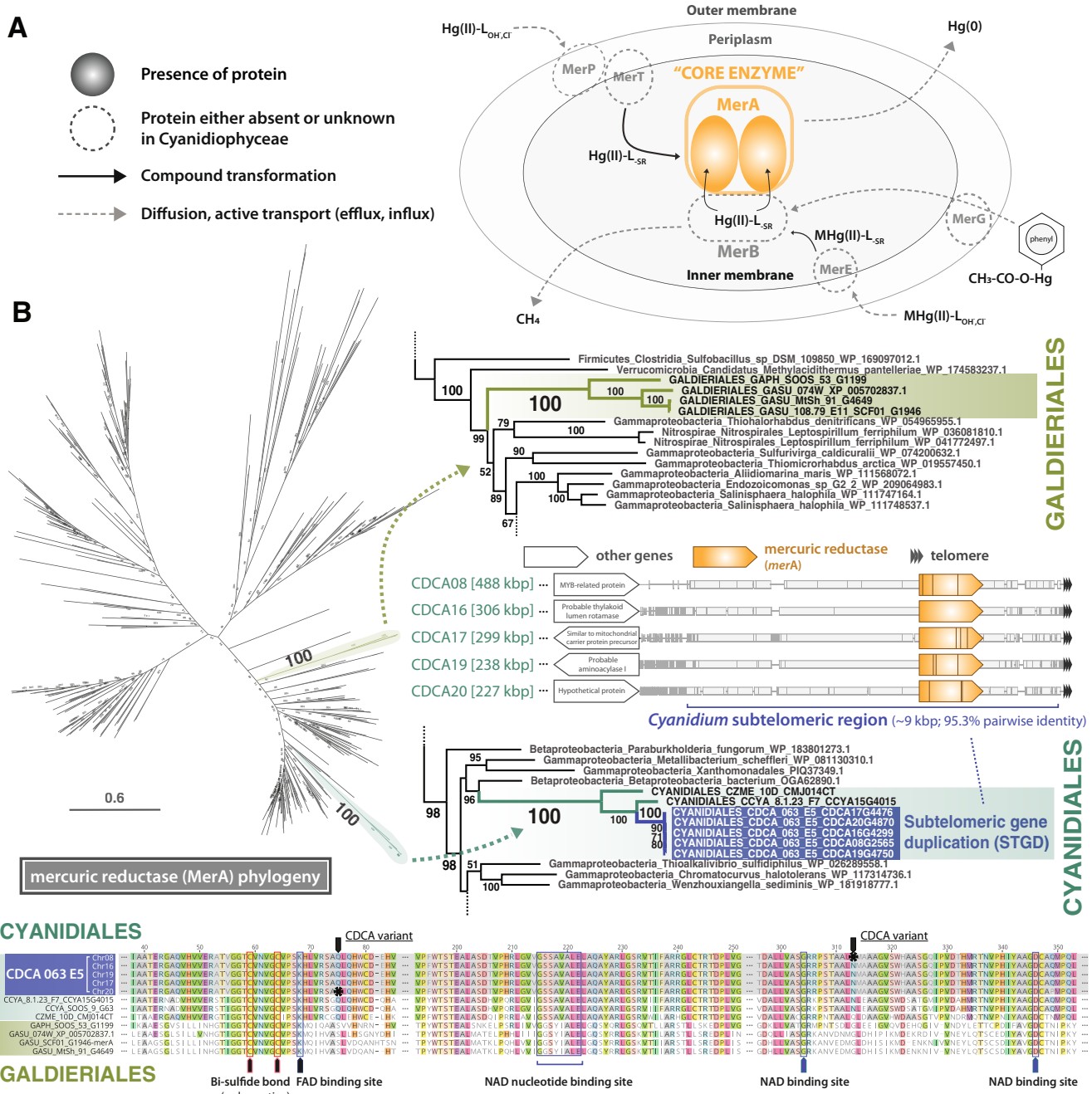

**Fig. 4 | Mercury detoxification in Cyanidiophyceae.** Highlighted text (turquoise) indicates subtelomeric gene duplications (STGDs). **A** Schematic diagram of modified mercuric detoxification pathway based on studies by Rojas and Boyd[45,115]. Only the core enzyme (i.e., *mer*A) was detected in cyanidiophycean genomes. MerA: mercuric reductase, MerB: organomercurial lyase, MerE: membrane protein that probably acts as a broad mercury transporter, MerG: periplasmic protein involved in cell permeability to phenylmercury, MerP: periplasmic mercury-binding protein, MerT: membrane mercury transport protein. **B** Maximum likelihood phylogeny of the *mer*A gene, with the protein sequence alignment. Based on the inferred tree, *mer*A genes were acquired independently by Cyanidiales and Galdieriales. Duplicated *mer*A genes were located at the subtelomeric termini of five chromosomes of *Cyanidium caldarium*.

in *Galdieria* are composed of unannotated proteins (hypotheticals) and transposable element-related genes such as retroelements and RNA-directed DNA polymerase (from the jockey mobile element). However, we also identified genes related to environmental adaptation. Putative archaeal-derived ATPases were found to be highly duplicated in *Galdieria* subtelomeric regions; these genes are linked to extreme habitats[14]. The existence of highly duplicated archaeal-derived ATPases suggests that this gene function was enhanced through subtelomeric duplications post-HGT (Supplementary Fig. 8b). Compared to other subtelomeric genes, several other

putative habitat-related genes (e.g., major facilitator superfamily, multidrug resistance protein, aluminum resistance protein) were duplicated in subtelomeric regions of *Galdieria* chromosomes (623/7021 [8.87%] genes in 14.5 Mbp of *Galdieria* 108.79 E11 genomes; see Supplementary Dataset 6). Although a few cases of recombination between subtelomeric regions have been reported from other eukaryotic lineages[34,35], gene expansion in Cyanidiophyceae is critical because of its highly reduced genome when compared to other free-living algae or eukaryotes[36]. STGDs may therefore provide a strategy for amplifying adaptive genes related to extremophily.

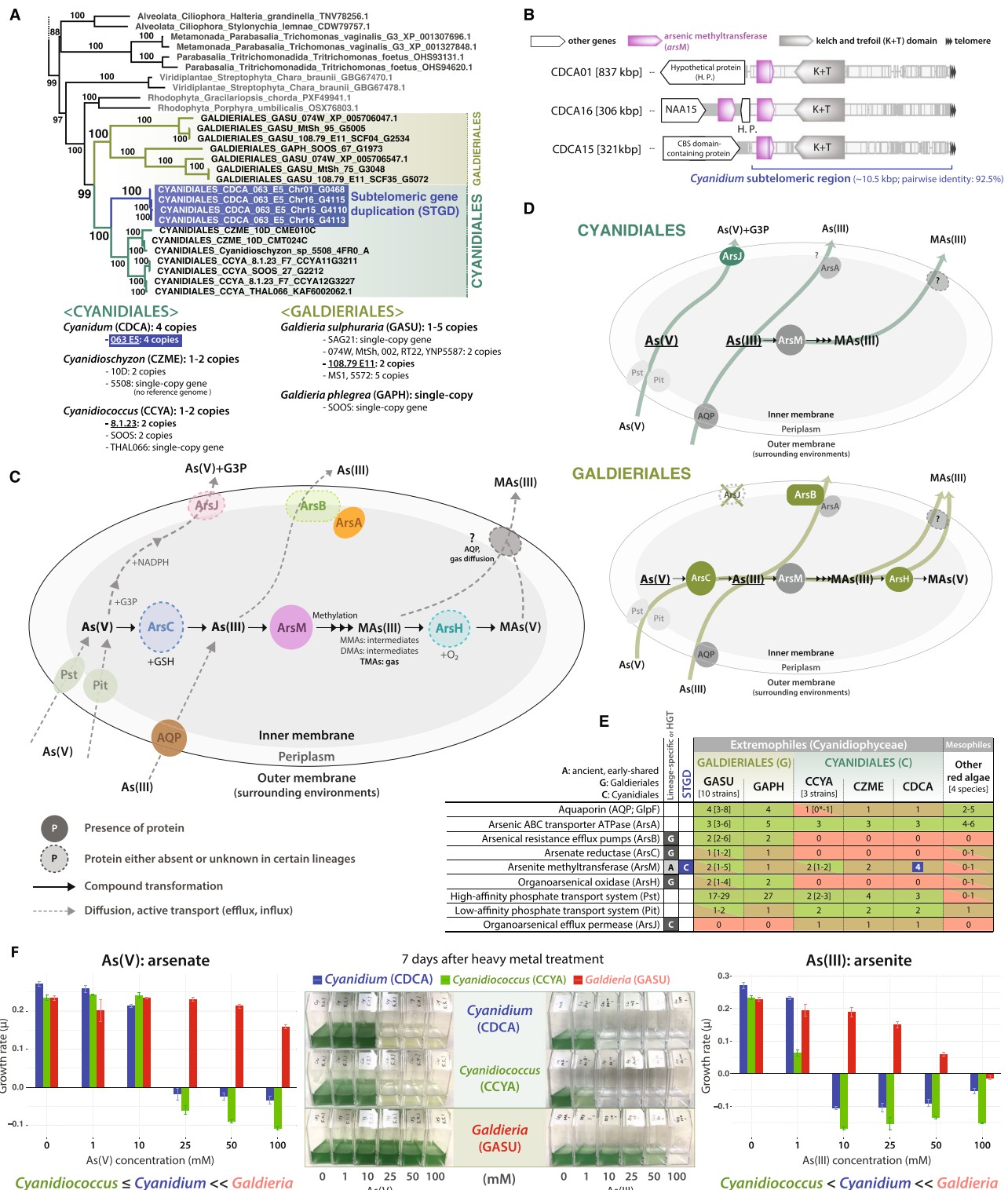

**Fig. 5 | Independent evolution of arsenic detoxification in Cyanidiophyceae.** Highlighted text (blue) indicates subtelomeric gene duplications (STGDs). **A** Arsenite methyltransferase (*ars*M) gene phylogeny in Cyanidiophyceae and a survey of the *ars*M gene in 16 Cyanidiophyceae. **B** Subtelomeric gene duplication of the *ars*M genes in the *Cyanidium* genome. **C** Schematic diagram of the modified arsenic detoxification pathway, based on Qin et al[52]. As(V): arsenate AsO$_4^{3-}$, As(III): arsenite AsO$_2^-$, MAs(III): methylarsenite (CH$_3$)$_n$-AsO$_2$, MAs(V): methylarsenate (CH$_3$)$_n$-AsO$_4$, G3P: glyceraldehyde-3-phosphate, GSH: glutathione. **D** Comparison of the arsenic detoxification pathway in Cyanidiales and Galdieriales. **E** Genomic survey of arsenic detoxification pathway genes in red algae. The red color

represents the absence of genes, the brown color represents a single copy of genes, and the green color represents more than two copies of genes. **F** Cell cultures of three cyanidiophycean species with different heavy metal concentrations. *Cyanidium* (CDCA) is represented by the color blue, *Cyanidiococcus* (CCYA) by the color green, and *Galdieria* (GASU) by the color red. 7 days after each heavy metal treatment, photos of culture flasks were taken. The growth rate (μ) was calculated using OD$_{750}$ differences between the initial culture and 7 days after heavy metal treatment. Bar plots with standard error bars were used to illustrate the mean growth rate that were calculated from triplicates (*n* = 3) in each concentration.

## Divergence of Cyanidiales and Galdieriales through extensive gene gain and loss events

To understand the trajectory of Cyanidiophyceae genome evolution, representative taxa from the major clades of Archaeplastida were chosen for orthologous gene family (OGF) analysis, whereby we considered both genome quality and evolutionary significance (see Supplementary Dataset 9). A total of 32,467 OGFs with 67,066 singletons were identified from ca. 380 K protein sequences from 26 representative species. We focused on lineage-specific gene gain and loss using Dollo parsimony[37]. Despite gene gain events (450 OGFs), most gene families show massive loss (1627 OGFs) during the divergence of red algae (branch 'a' in Fig. 3) from its non-photosynthetic sister group, *Rhodelphis*[38]. Excluding unidentified genes from clusters of orthologous genes (COGs), the major functional category of gene gain was of 'O: posttranslational modification, protein turnover, chaperones', which contains 31 gene families, but other COGs were detected as much as the top matched COG (e.g., 22 OGFs of 'T: transcription', 24 OGFs of 'U: intercellular trafficking, secretion, and vesicular transport') (Supplementary Fig. 13). The massive gene loss in ancestral red algae was primarily related to 'T: signal transduction mechanisms' (229 OGFs) and this event caused flagella (e.g., IFT-A and IFT-B genes) and basal body degeneration, loss of glycosyl-phosphatidylinositol (GPI) anchor biosynthesis, and autophagy[36].

Following the first massive gene loss event in the red algal ancestor, the second loss event (1282 OGFs lost) occurred in the ancestor of Cyanidiophyceae (branch 'b' in Fig. 3). These losses primarily impacted 'O: posttranslational modification, protein turnover, chaperones' (branch 'b' loss in Supplementary Fig. 13). One of the key events in Cyanidiophyceae evolution was the loss of *Dicer-like RNase III endonuclease 1* (*DCL1*) and *ARGONAUTE 1* (*AGO1*), which are essential components of the microRNA (miRNA) processing pathway and miRNA-mediated gene silencing, respectively[39,40]. In contrast to the gene losses in Cyanidiophyceae, only a small number of gene families (66 OGFs) were gained by the ancestor of this lineage. However, major gene gains were found to have occurred independently during the diversification of Cyanidiales (621 OGFs) and Galdieriales (494 OGFs) along with independent gene losses (−911, −507, respectively) (see branch 'c' and 'd' in Fig. 3). These independent gene gain/loss events resulted in gene number differences (ca. 1.0–2.5 K genes) between the two lineages. For instance, reduction of the spliceosomal machinery in Cyanidiales drove (or were driven by) intron loss in Cyanidiales genomes (e.g., 36 introns in CCYA 8.1.23 F7, 46 introns in CDCA 063 E5, 27 introns in CZME 10D) (Table 1)[41], whereas Galdieriales largely preserved the spliceosome, resulting in intron-rich genes (>10 K introns). The acquisition of archaeal ATPase (adenosine triphosphatases) genes was one of the major events in Galdieriales evolution (branch 'd' in Fig. 3), which may reflect an adaptation to temperature fluctuations[14]. In addition, previous cyanidiophycean genome studies have demonstrated that functions of the majority of HGTs (96 genes) in Cyanidiophyceae (particularly *Galdieria* spp.) are related to polyextremophilic adaptations (e.g., metal and xenobiotic resistance/detoxification, cellular oxidant reduction, carbon and amino acid metabolism, osmotic and salt tolerance)[9,14,15]. Consequently, many lines of evidence demonstrate a functional correlation between HGTs and adaptation to extreme environments.

Highly diverged genomic features between Galdieriales and Cyanidiales species also likely resulted in phenotypic differences (e.g., size, shape, and organelle features) and local adaptation to microhabitats[19]. Galdieriales occupies a more diverse variety of niches in extreme environments (e.g., mine drainage sites, endolithic environments) than do Cyanidiales species, whose habitats (e.g., ditches and streams near hot springs) may be more ecologically stable[13,42]. Cyanidiophyceae lineages have therefore spread to different extreme microhabitats that have led to divergent patterns of genome evolution, even at the species level, where minor variations also presumably reflect the occupied niche.

## Heavy metal resistance via horizontal gene transfer and subtelomeric gene duplication

The pattern of STGDs is lineage specific. For instance, some duplicated subtelomeric genes in *Cyanidium* are associated with environmental adaptation, which is linked to heavy metal resistance. Cyanidiophycean species thrive in thermoacidic habitats (e.g., Yellowstone National Park) with high arsenic (As: -3.57 mg/L)[43] and mercury (Hg: -710 µg/L) concentrations[44]. Mercuric reductase (*mer*A) is a central enzyme in mercury detoxification (*mer*) that catalyzes the reduction of Hg(II) to the less toxic (i.e., reactive) volatile Hg(0) (Fig. 4A)[45]. In contrast to the *mer* operon in Bacteria, that includes additional accessory proteins, the *mer* system in Archaea is solely based on a *mer*A gene[46]. The broadly sampled MerA phylogeny shows that the *mer*A gene originated in a thermophilic bacterium after the divergence of Archaea and Bacteria, and subsequently was acquired in Archaea through HGT[46]. *Mer*A genes have been identified in all cyanidiophycean genomes but not in mesophilic red algae or other Archaeplastida. Phylogenetic evidence suggests that the Cyanidiophyceae *mer*A gene was derived from Bacteria via HGT (Fig. 4B). Furthermore, Galdieriales and Cyanidiales show paraphyly in the *mer*A gene phylogeny, implying that these two lineages may have acquired the *mer*A gene through independent HGT events (Fig. 4B). Interestingly, *mer*A genes in *Cyanidium* underwent duplication resulting in five copies, all of which genes are found in subtelomeric regions. This differs from other Cyanidiales species that contain a single *mer*A gene copy in a non-subtelomeric region. This result suggests that *mer*A genes were amplified via subtelomeric duplication in *Cyanidium*. Because there were no *mer* operon-related genes in the genomes (e.g., *mer*R: mercury-dependent transcriptional regulatory gene, *mer*B: organomercurial lyase gene, *mer*T: membrane mercuric transporter gene), which are typically found as accessory proteins in other eukaryotes and Bacteria and are required for mercury detoxification[47], the *mer*A gene may act alone in this process in Cyanidiophyceae. It is likely that Hg(0) is excreted in these algae through an ancestral-derived transport system (e.g., multidrug resistance protein [ABC transporter G family])[48] or alternatively, through diffusion (Fig. 4B).

The arsenic detoxification pathway proceeds by excreting mono-(M-), di- (D-), and tri-methylated (T-) arsenic metabolites (e.g., MAs(III), MAs(V)) produced by a multistep process[49]. Arsenite methyltransferase (ArsM; AS3MT; SAM) is the key arsenic detoxification enzyme that methylates arsenic compounds and has a complex evolutionary history with multiple HGT events in eukaryotes[49–51]. Although an ancient eukaryotic HGT has previously been identified[50], cyanidiophycean *ars*M genes share a common ancestry with other red algae (Fig. 5A). When ArsM was compared to mesophilic red algae, Cyanidiales ArsMs (e.g., CmArsM7, CmArsM8) were more thermotolerant ($T_{opt}$ of 60–70 °C) with vicinal cysteines that could serve as strong As(III) binding sites[52]. In addition, *ars*M genes have undergone independent gene duplication in each cyanidiophycean species (1–4 copies; Fig. 5A). For example, another subtelomeric duplication was found in the *Cyanidium* genome, where *ars*M genes were positioned near kelch domains within the subtelomeric regions (Fig. 5A, B). This implies that the integrated *ars*M and kelch gene regions were duplicated together in this species. Other cyanidiophycean *ars*M genes (1–4 copies) were not detected in the subtelomeric region. The copy number of *ars*M gene in Cyanidiales may explain the different As(III) tolerances among these species, with the greater tolerance being found in *Cyanidium* compared to *Cyanidiococcus* (Fig. 5F).

After identification of the *ars*M gene duplications, we analyzed the arsenic pathway based on the enzymatic mechanism described from a *Cyanidioschyzon* arsenic transformation study as well as a few other arsenic detoxification pathway studies[51–54]. We inspected these genes

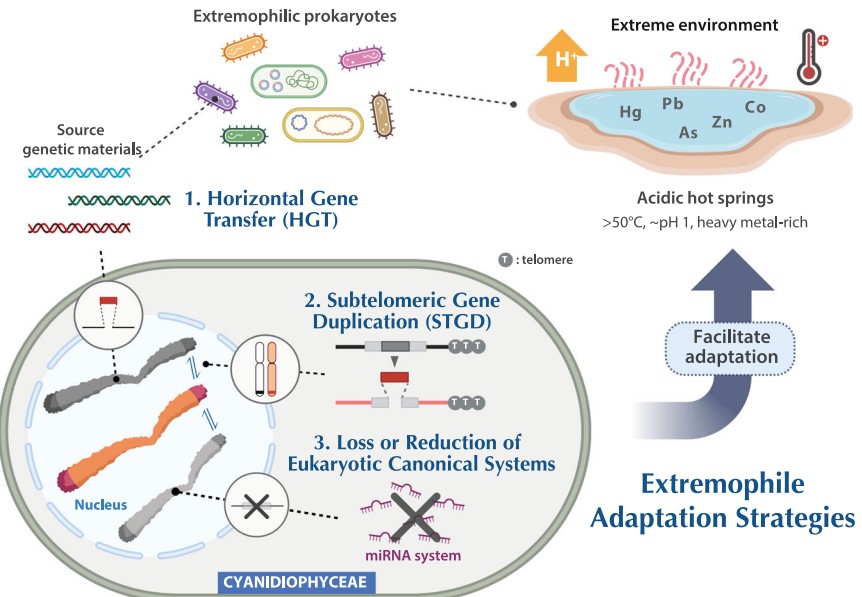

**Fig. 6 | Schematic model of the adaptation to extremophilic lifestyles in Cyanidiophyceae.** Cyanidiophycean algae successfully adapted to extreme environments via the three following processes: 1) horizontal gene transfer (HGT), 2) subtelomeric gene duplication (STGD), and 3) loss or reduction of eukaryotic canonical traits. BioRender was used to create the graphical summary.

in other Cyanidiophyceae species and used transit peptide prediction and transmembrane region prediction to confirm their possible localization. Surprisingly, we found metabolic pathway differences among the lineages, even in thermoacidic Cyanidiales and Galdieriales (Fig. 5D). Some arsenic-related transporters derived from the eukaryotic ancestor (e.g., arsenic ABC transporter ATPase, aquaporin, high & low-affinity transport system) show significant differences in their copy number (Fig. 5E). For instance, the Pst (high-affinity inorganic transporter) gene showed >4-fold copy number difference (2–4 copies in Cyanidiales, 17–27 copies in Galdieriales). Most of these genes were likely derived from the red algal common ancestor, however the arsenite efflux pump (*ars*B) and arsenate reductase (*ars*C) genes that oxidizes As(III) to As(V) only exist in Galdieriales and were acquired via HGT (Fig. 5C)[14,15]. Due to the presence of the *ars*C and *ars*B genes in *Galdieria* species, our arsenate tolerance experiment showed a greater tolerance to As(III) and As(V) in two *Galdieria* species than in *Cyanidium* and *Cyanidioschyzon*, which lack the ability to oxidize As(III) (Fig. 5D, F). The growth rate results indicate that integrating the As(III) and As(V) pathways enabled Galdieriales to develop a more efficient arsenic detoxification system than that of the Cyanidiales, which has separate detoxification pathways for As(III) and As(V). Another interesting result is that the *ars*J gene was only found in Cyanidiales. This organoarsenical efflux permease gene (*ars*J) is a member of the MFS transporter family that is involved in As(III) efflux. The *ars*J genes exist as a gene cluster (ArsJ-GAPDH-PGK) in green algal lineages[55]. The *ars*J gene exists by itself in Cyanidiales, but the *ars*J phylogeny shows monophyly with Viridiplantae and eukaryotic lineages, implying that Cyanidiales species retained this gene to detoxify arsenite (Supplementary Fig. 14). Taken together, we conclude that multiple mechanisms, including: 1) STGDs (e.g., *ars*M genes in *Cyanidium*), 2) HGTs (*ars*B and *ars*C in *Galdieria* species), and 3) independent gene losses (*ars*J), led to the evolution of the arsenic pathway in Cyanidiophyceae and resulted in lineage-specific differences in the ability to tolerate various arsenic concentrations (Fig. 5F). Our experimental results from arsenite and arsenate heavy metal treatment experiments show that *Galdieria* has higher tolerance to these metals than other Cyanidiales species. This may be related to their microhabitat, because *Galdieria* species inhabit a broader range of environments (e.g., endolithic) than do Cyanidiales species, which are found in a narrower range of more

ecologically protected niches (e.g., hot springs)[13]. In *Galdieria* species habitats, heavy metal concentrations may fluctuate due to the evaporation of humidity in exposed environments, necessitating greater tolerance to heavy metals.

## Loss of the miRNA system in Cyanidiophyceae

The miRNA system is required for transcriptional and post-transcriptional gene silencing, both of which are important for controlling the expression of protein coding genes during development or in response to environmental cues[56]. In plants, RNA polymerase II (Pol II) transcribes *MIR* to produce primary miRNAs (pri-miRNAs), which are then processed into miRNA/miRNA* duplexes by DCL1 and its associated proteins[57]. AGO1 then selects the miRNA strand to form the RNA-induced silencing complex (RISC), which targets transcripts that are either cleaved or translationally suppressed[58]. Both DCL1 and AGO1 homologs have been reported from mesophilic red algae (i.e., *Gracilariopsis chorda*), but not from *Cyanidioschyzon*[59,60]. We found that those two key regulators of small RNA metabolism were lost in all Cyanidiophyceae (Supplementary Note 4; Supplementary Fig. 15; Supplementary Dataset 10). Specifically, we searched for DCL and AGO genes from representative species of Archaeplastida and found them in most taxa excepting Cyanidiophyceae and the marine oligotroph, *Ostreococcus tauri* (Supplementary Fig. 15a). The miRNA system is anciently derived and has been secondarily lost in some lineages (e.g., yeast: *Ustilago maydis*), implying that it is not necessarily needed for survival[61,62]. Although the miRNA system is missing in Cyanidiophyceae, we provide evidence for the putative existence of other epigenetic regulatory mechanisms such as long-noncoding RNAs (Supplementary Note 5; Supplementary Fig. 16), DNA methylation, and histone modification, including the polycomb group in Cyanidiophyceae (Supplementary Note 6; Supplementary Fig. 17).

Due to the various approaches used to interpret genome evolution (e.g., size, gene content), interpreting gene losses due to "evolutionary pressure" can be controversial with different competing explanations, such as: i) the mutational hazard hypothesis (genetic drift), ii) the nucleotypic and nucleoskeletal hypotheses, and iii) the genome streamlining hypothesis (natural selection)[63]. With regard to the genome streamlining hypothesis, studies have shown that stressful environments (e.g., nutrient-limited) can result in evolutionary

pressure to reduce energy or material costs through genome streamlining both in eukaryotes and in prokaryotes[64–67]. Certain evolutionary constraints would result in genome reduction or gene loss (environment-dependent conditional dispensability) in these cases[68]. Thus, we propose that the strong evolutionary constraints imposed by external factors (e.g., heavy metal exposure, thermal stress) resulted in the parallel loss of functionally equivalent genes.

## Extremophilic adaptation of proteins

Compositional change in proteins adapted to thermophily is an interesting aspect of Cyanidiophyceae evolution. Previous analyses have shown that proteins such as arsenic methyltransferase (CmArsM7) from *Cyanidioschyzon* sp. 5508 to have a temperature optimum at 60–70 °C[52]. Analysis of reference Cyanidiophyceae proteomes also show differences in features such as aggregation, when compared to mesophilic red algae or other lineages (Supplementary Note 7; Supplementary Figs. 18, 19). This result corroborates data from Galdieriales mitochondrial proteins that indicate protein property changes as a key evolutionary transition that facilitated thermoacidophilic adaptation of Cyanidiophyceae[19].

## Other unique features of Cyanidiophyceae

It is intriguing to note how many differentiating characteristics of Cyanidiophyceae have resulted from genomic adaptation to extreme environments. These include streamlined genomes, adaptive HGT, reduced spliceosomal activity (absent in prokaryotes), and polyextremophily[9,14,15]. Another unique trait present in some Cyanidiophyceae is expansion of the polycistronic gene expression system; about 14.5% of genes in *Cyanidium* display this feature (Supplementary Note 8; Supplementary Fig. 20; Supplementary Dataset 11; a list of identified proteins encoded by polycistronic transcripts are provided in Dryad database). Therefore, despite inhabiting different domains of life, persistence in harsh, hot springs environments has wrought a similar set of adaptations that allow Cyanidiophyceae to thrive alongside prokaryotes, with which they compete for precious resources.

## Three major extremophile adaptation strategies

This study elucidates three major drivers (horizontal gene transfer [HGT], subtelomeric gene duplication [STGD], and gene/genome reduction) of Cyanidiophyceae genome evolution that have allowed these taxa to adapt to polyextreme environments. Specifically, prokaryotic genes obtained through HGT provided benefits to Cyanidiophyceae with respect to heavy metal detoxification, and some of those genes were amplified via STGD. The pattern of STGDs across shallower and deeper taxonomic levels demonstrates that these events reflect local adaptation to specific microhabitats occupied by (often) neighboring Cyanidiophyceae. Other studies of subtelomeric regions from different lineages such as *Trypanosoma* (Euglenozoa), *Plasmodium* (Apicomplexa), and *Candida* (Fungi), show that their rapid evolution leads to the origin of a large repertoire of genes that confer selectively beneficial characteristics[69]. As a result, using various combinations of the extremophile adaptation strategies, Cyanidiophyceae successfully adapted to diverse microhabitats, resulting in a unique lineage of photosynthetic eukaryotes that have thrived in extreme environments for more than 1 billion years (Fig. 6).

# Methods

## Sample preparation

To eliminate mixed cryptic species in a culture strain (e.g., Supplementary Fig. 21), we established cultures from cells using the fluorescence-activated cell sorting (FACS) method: *Cyanidium caldarium* 063 E5 was isolated from DBV 063 strain, *Cyanidiococcus yangmingshanensis* 8.1.23 F7 was isolated from *Galdieria maxima* (now *Cyanidiococcus yangmingshanensis*) 8.1.23 strain, and *Galdieria*

*sulphuraria* 108.79 E11 was isolated from SAG 108.79 strain (Supplementary Fig. 22). After initial cultivation in 96 well plates, the mass culture was done in modified 5x Allen's medium (Supplementary Dataset 12). Total genomic DNAs and RNAs were extracted using a modified cetyltrimethylammonium bromide (CTAB) method and RNeasy Plant Mini Kit (Qiagen, Hilden, Germany) following its protocol, respectively.

## Whole genome sequencing (WGS) and whole transcriptome sequencing (WTS)

For genome and transcriptome sequencing, both short-read and long-read sequencing were conducted (Supplementary Dataset 1). For PacBio whole genome sequencing (WGS), we used SMRTbell® Express Template Prep Kit 2.0 (Pacific Biosciences, Menlo Park, CA, USA) with a 15 kbp size selection to construct Sequel I sequencing libraries of *Cyanidium* and *Cyanidiococcus*. For *Galdieria* PacBio WGS, SMRTbell® Express Template Prep Kit 1.0 (Pacific Biosciences) with a 9 kbp size selection was used to prepare the RS II sequencing library and SMRTbell Express TPK 2.0 (Pacific Biosciences) was used for HiFi library preparation. All experiments followed the manufacturer's standard protocol, without shearing step in *Cyanidium* and *Galdieria* samples. SQK-LSK109 ligation kit (Oxford Nanopore Technologies, Oxford, UK) was used to construct a library of *Galdieria* PromethION sequencing without shearing step and a 20 kbp size selection. For Illumina HiSeq2500 WGS of *Cyanidium* and *Galdieria* species, TruSeq® Nano DNA Prep Kit (Illumina, San Diego, CA, USA) with an insert size 550 bp was used to prepare gDNA sequencing libraries. The same kit and protocol were used for *Cyanidiococcus* WGS and ran in the Illumina NovaSeq6000 platform. SMARTer PCR cDNA Synthesis Kit (Clontech Laboratories, Palo Alto, CA, USA) and SMRTbell® Express Template Prep Kit 1.0 (Pacific Biosciences) were used to prepare PacBio WTS (Iso-Seq) libraries. Clustering and deduplication of Iso-Seq reads were done by IsoSeq v3 implemented in Sequel SMRT® Link v8.0 and high-quality reads (99% accuracy) from clustered results were only used for subsequent analysis. For Illumina WTS (RNA-Seq), TruSeq® Stranded mRNA Prep Kit (Illumina) were used for library construction for all species and those libraries were sequenced with Illumina NovaSeq6000 platform. Adapter and quality trimming for Illumina sequencing reads were conducted using Trimmotatic v0.36[70] with parameter settings of 'ILLUMINACLIP:TruSeq3-PE.fa:2:30:10:2:-keepBothReads LEADING:3 TRAILING:3 MINLEN:100'.

## Genome size estimation and genome assembly

We chose different approaches for genome assembly of individual species due to the differences in sequencing methods and assembly performances. Although the basic outline of the assembly process is consistent across species, we used multiple platforms and methods to improve the quality of each species' assembly. The basic outline of assembly is as follows: i) build a draft assembly using long-read sequencing platforms (e.g., PacBio, Nanopore) applying multiple assemblers (e.g., HGAP, CANU, FALCON, MaSuRCA), ii) sort out organelle genomes (e.g., mitochondria, chloroplast) to get nuclear genome assembly only, iii) use additional scaffolding method (e.g., RaGOO) based on reference assembly or manually complement non-covering regions from other assemblers, iv) use haplo-merging tools (e.g., Purge-Dups, Purge Haplotigs) to remove duplicated regions that are not considered necessary in a haploid genome, v) correct assembled genome using Illumina reads (e.g., Bowtie2, Pilon) and assess chimeric region based on mapping coverage of reads (Supplementary Fig. 23).

For *Cyanidium caldarium* 063 E5, we used two assemblers, MaSuRCA v3.4.2[71] was used for the main genome and miniasm v0.3-r179[72] was used for complementing regional differences between two assembled contigs. RaGOO v1.1[73] with long read validation was used for contig scaffolding to finalize scaffolds and Purge-Dups v1.2.5[74] was used to remove haplotigs. Finally, 20 chromosomes were recovered

from the scaffolding process, and chromosome sequences were processed for error correction with pre-processed short-read data using Bowtie2 v2.3.4.1 ('very-sensitive' option)[75] and Pilon v1.23[76]. We repeated this correction step until no conflict sequence was found between corrected and query genomes.

The draft genome of *Cyanidiococcus yangmingshanensis* 8.1.23 F7 was assembled using HGAP4[77] as suggested in the PacBio SMRT portal and we compared the result with the FALCON-Unzip v1.8.1[78] assembly. Organelle genomes were separated from assembled genomes using previously established plastid genomes and mitogenomes[19]. We were able to recover 20 chromosomes from HGAP4 without using any scaffolding process, and FALCON contigs were used to refine the subtelomere regions. Genome correction, as done in *Cyanidium*, was used to fine-tune the genome sequence after recovering *Cyanidiococcus* chromosomes.

Because hybrid assembly using different platform sequencing did not result in a robust assembly of the *Galdieria* genome, we used different combinations of data for assembly; i) FALCON assembler v0.3.0 using PacBio HiFi reads, ii) Nanopore sequencing-based CANU v2.2 assembly, iii) PacBio RS II-based HGAP3, and iv) MaSuRCA v3.2.4 (PacBio and Illumina hybrid assembly). The basic structure of the *Galdieria sulphuraria* 108.79 E11 genome was built using HiFi result, and other assemblers were used for genome scaffolding and obtaining unique gene regions that the HiFi assembly did not cover. Because the *Galdieria* genome has higher heterozygosity than other Cyanidiales lineages and shows a diploid signal, we used different correction tools (e.g., Pilon v1.2.4, NextPolish v1.2.3, Hapo-G v1.0) using Illumina and PacBio HiFi reads with multiple replications for genome polishing[76,79,80]. In addition, due to small chromosome sizes and duplicated regions across chromosomes, it was challenging to discriminate, or pair each chromosome to generate a haploid genome. As a result, we decided to include a few overlapping chromosomal contigs (e.g., haplotigs) in the *Galdieria* genome under a pan-genome concept.

PacBio reads were mapped to assembled genomes using minimap v2.17-r941[72] after all the genomes were reconstructed, and WGSCoveragePlotter[81] was used to visualize mapping coverage of each species (Supplementary Fig. 23). We used Jellyfish v2.2.8[82] and KMC v2.3.0[83] to count *k*-mers and estimated genome size using Genome-Scope 2.0[84]. When compared to the estimated genome size using *k*-mers, the assembled genome covered at least 90% of the predicted size (Supplementary Fig. 24).

## Gene modeling and annotation

After reconstruction of genomes, we mapped Illumina RNA-Seq and PacBio Iso-Seq data by STAR(long) v2.7.5a[85] to identify transcribed regions from genome data. Transcriptome-mapped data (e.g., Illumina RNA-Seq, PacBio Iso-Seq) was used for the training set of ab initio gene modeling and BRAKER v2.1.6[86] and GeMoMa v1.7.1[87] were used for gene annotation. However, unlike *Galdieria* species, BRAKER-based gene annotation did not work well with Cyanidiales genomes due to Cyanidiales unique gene features (e.g., intron-poor gene, short intergenic region). Considering these features, we used Augustus v3.3.1[88] for ab initio modeling based on BUSCO training sets and exonerate v2.4.0[89] for homology-based gene prediction using reference proteins of *Cyanidioschyzon* and *Cyanidiococcus*[21,29]. Combining all gene modeling results with RNA-Seq and Iso-Seq mapping information, we finalized and corrected gene modeling by manual inspection of integrated information (e.g., ab initio gene modeling, reference proteome homology-based gene modeling, transcript-mapped regions) in all three species. Additionally, some of the putatively mispredicted genes in the *Galdieria* genome (approximately 70 genes) were manually removed based on two criteria: i) exclusive intron patterns without support from RNA-seq and Iso-Seq data, ii) no homology with other proteins and lack of a function domain inside the protein. The completeness of gene modeling was verified by BUSCO v3.0.2 using

the general eukaryote database ('eukaryota_odb9')[90]. Despite the availability of a more recent BUSCO database ('eukaryota_odb10, *n* = 255'; 21.1% missing BUSCOs in *Cyanidioschyzon* 10D), we chose to use previous version database ('eukaryota_odb9', *n* = 303'; 3.6% missing BUSCOs in *Cyanidioschyzon* 10D) because recent version contains many missing genes that were lost in the cyanidiophycean lineage tested by reference genome (*Cyanidioschyzon* 10D). We used multiple methods for functional annotations of genes in each species: i) MMSeqs2-based search against NCBI nr protein database, ii) HMMER-based search against a customized HMM database of KEGG orthologs using KofamKOALA (ver. 2021-03-01)[91], iii) DIAMOND-based search using eggNOG v5.0[92], which is specialized database for functional annotation. For functional RNA annotation, we applied Infernal v1.1.2[93] using Rfam v12.5 (March 2021, 3940 families)[94] database. Transcription start site prediction was identified by TSSPlant[95] with the support of in-house python script.

Repeat sequences in genomes were identified using the de novo method in RepeatModeler v2.0.2a (http://www.repeatmasker.org/RepeatModeler) following the analysis pipeline used in a previous study[25]. We used 13 and 14 *l*-mers optimized from 'log4[genome size] + 1' for the repeat analysis and classified them into repeat subclasses using RepBase (updated October 26th, 2018) and Dfam v3.3 (November 09th, 2020) database. The genetic distance between repeat copies found were extracted from the output of RepeatMasker v4.1.2-p1 and used to calculate Kimura distance values[96].

## Circular dichroism (CD) spectroscopy

Oligonucleotides were designed based on telomeric repeats in cyanidiophycean species and compared to previously confirmed G-quadruplex forming telomeric tandem repeats (Supplementary Dataset 2)[97]. DNA samples for CD spectroscopy were prepared in 10 mM Tris−HCl (pH 7.4), 1 mM EDTA, 150 mM KCl, and 40% (w/v) PEG 200 cat. P3015 (Sigma-Aldrich, St Louis, USA) to induce the macromolecular crowding effect and stabilize G-quadruplex structures. Before the experiment, the DNA mixtures were heated at 95 °C for 5 minutes and cooled to room temperature (at least 20 min). Circular dichroism (CD) measurements of oligonucleotides were performed on a Jasco J-815 spectropolarimeter (Jasco, Tokyo, Japan) at 25 °C using Hellma® Macro-cuvette 110-QS (1 mm path length). CD spectra of various DNA samples (5 µM DNA) were recorded from 350 to 200 nm using a 1 nm scale and a scanning speed of 100 nm/min. CD spectra measurements were repeated three times for each sample, and mean values were used. The 'ggplot2' R package's 'geom_smooth' function was used to plot CD spectra (mdeg) by wavelength (nm).

## Genome analysis

Nucleotide sequence alignment-based genome comparisons were done using JupiterPlot v1.0 (https://github.com/JustinChu/JupiterPlot) to identify structural variation. However, nucleotide alignment-based genome comparison between cyanidiophycean species had insufficient resolution, so we did gene synteny-based comparison for higher levels of taxonomy. Genomes were compared using synteny blocks identified by MCScanX[98] with a minimum syntenic block length of five genes and a maximum gap between genes in a syntenic block of 25 genes[25]. Tree view mode of SynVisio (https://synvisio.github.io/) was used to visualize the results of the synteny block comparison.

The grouping of orthologous genes was performed by Orthofinder v2.5.2[99] with default option[100] and protein dataset were collected from 36 representative taxa of Archaeplastida (Supplementary Dataset 9, see Dryad database for orthogroup information). Gene gain and loss events of cyanidiophycean algae were tested by the Dollo parsimony method (DolloP) using Archaeplastida-based orthogroups[25,101]. We used this orthogroup information for downstream analysis of gene families, however, two major issues arose: i) some misannotated genes found in individual strains combined two independent gene families

into one orthogroup that has no functional domain in common but is clustered together by a misannotated fused gene, and ii) some orthogroups were separated due to protein properties (e.g., protein divergence, size) due to a unified parameter adjusted for all different gene families. We were not able to discard some problematic genes from whole orthogroups because we do not have strong evidence to reject published gene modeling data. Therefore, we manually confirmed unvalidated genes that appeared to be misannotated compared to sister species or strains (i.e., parsimonious approach) for further analysis. TargetP v1.1[102] and DeepTMHMM v1.0.1[103] were used to predict transit peptides and transmembrane domain regions in order to validate gene localization.

## Phylogenetic analysis of genes

To determine the evolutionary history of target genes, we obtained homologous protein sequences from the NCBI non-redundant protein sequence database by using protein similarity searches with MMSeqs2 v13.45111[104]. Sequences collected for phylogenetic analysis were aligned using MAFFT v7.310, and some alignments containing many gaps were trimmed using trimAl v1.4 '-automated1' option[105]. IQ-TREE v2.1.2 was used for Maximum Likelihood (ML) inference of phylogenetic tree[106]. To select evolutionary models, implemented model selection was used, and ultrafast bootstrap approximation approaches (1000 replications, UFBoot2) were used for phylogenetic analysis[107]. After phylogenetic tree reconstruction, we removed taxa that appeared to be redundant due to issues with taxon sampling (e.g., extensively sequenced in a particular lineage). Following the removal of redundant taxa, we reanalyzed the datasets, beginning with the alignment and performing the phylogenetic analysis, as described above. The final trees were visualized using FigTree v1.4.4 (https://github.com/rambaut/figtree) with a midpoint root or an unrooted tree if outgroups were not considered from the start.

## Identification of subtelomere and gene duplication ratio

To identify subtelomeric regions from genomes, LASTZ alignment v7.0.2 was used to determine if there were conserved regions between chromosomes[108]. Subtelomere regions near telomeric repeats were manually confirmed using LASTZ alignments across chromosomes, and subtelomeric genes were identified within subtelomeric regions (Supplementary Fig. 8).

We attempted to remove paralogs from gene duplication detection and focus on recent gene duplications in order to calculate the proportion of subtelomeric gene duplication when compared to the overall number of gene duplications. DIAMOND v2.0.5.143 with variable parameters was applied to conduct protein homology searches (blastp) between each protein sequence in the entire proteomes. Query and subject coverage were set to 70 to 90% with 5% intervals, and protein identity was set to 70–90% with 5% intervals as well. As a result, this analysis used a total of 25 parameter combinations, which were visualized in a plot (Supplementary Fig. 10; Supplementary Dataset 13). Fisher's exact test ('fisher.test'), implemented in R was used independently to test the significance of subtelomeric regions and gene duplication in each species.

## Non-synonymous substitutions per non-synonymous sites ($K_a$) and synonymous substitutions per synonymous sites analysis ($K_s$)

To assess selection acting on subtelomeric duplicated genes, each subtelomeric duplicated gene was aligned using MAFFT v7.471[109]. $K_a/K_s$ analysis were done using ParaAT v2.0[110] and KaKs_Calculator v2.0[111].

## Analysis of histone modification ChIP-Seq data

We used previously sequenced ChIP-Seq data (input DNA, histone H3 [H3] and tri-methylation of lysine 27 on histone H3 [H3K27me3]) from *Cyanidioschyzon merolae* 10D to confirm the H3K27me3 histone modification pattern in Cyanidiophyceae[33]. All ChIP-Seq data were mapped against the *Cyanidioschyzon* genome using Bowtie2 v2.3.4.1[75], and peaks were identified with Model Based Analysis of ChIP-seq data (MACS3 v3.0.0a7)[112]. Input DNA data were used as a control for both H3K27me3 and H3. Enrichment of H3K27me3 peaks refer to the MACS3-calculated log fold changes over H3 and we used calculated fold-enrichment information for further analysis. We used IGV v2.11.0[113] for visualizing the output findings of "broadPeak" and "gappedPeak" which were signal enrichment based on pooled and normalized data.

## Heavy metal treatments

A modified Allen's medium with increasing concentrations of each metal (0, 1, 10, 25, 50, and 100 mM, pH=2) was used to test the arsenite (As(III); NaAsO₂, CAS #7784-46-5, Sigma-Aldrich) and arsenate (As(V); Na₂HAsO₄·7H₂O, CAS #10048-95-0, Sigma-Aldrich) tolerance of *Cyanidiococcus yangmingshanensis*, *Cyanidium caldarium*, and *Galdieria sulphuraria*. Physiological experiments with three biological replicates were conducted with a shaking speed of 130 rpm at 30 °C and a light intensity of 70 μmol/m²·s at a 12:12 h light-dark cycle for 7 days. On the first day, cell density was diluted to $OD_{750}$ as 0.05 to standardize the initial condition. $OD_{750}$ was measured using xMark™ Microplate Absorbance Spectrophotometer (Bio-Rad, Hercules, USA) on the first and the seventh days of the experiment. For growth rate (μ) calculation, we used the corrected $OD_{750}$ value (sample $OD_{750}$-blank $OD_{750}$), and the following equation was applied:

$$\mu = \frac{\ln OD_2 - \ln OD_1}{(t_2 - t_1)} \tag{1}$$

(where $\mu$ indicates the growth rate per day, $OD_n$ indicates corrected $OD_{750}$ value of measured point, $t_n$ indicates the number of days after heavy metal treatment).

## Characterization and verification of polycistronic transcripts

We used deduplicated high-quality transcripts from PacBio Iso-Seq circular consensus sequencing (CCS) reads to identify polycistronic transcripts, and all transcripts were mapped to the genome using STARlong v2.7.5a[85]. Using gene modeling information and mapped information, polycistronic transcripts were identified using an in-house python script based on the criterion of complete coverage of at least two gene regions in the same direction as the transcript. After identifying polycistronic loci, internal ribosome entry sites (IRESs) were determined from all putative polycistronic transcripts using IRESfinder[114].

To verify polycistronic gene expression, we synthesized cDNA using Thermo Scientific First Strand cDNA Synthesis Kit cat. #K1612 (Thermo Fisher Scientific, Waltham, USA). Before synthesizing cDNA from extracted RNAs, we treated DNase I cat. #EN0521 (Thermo Fisher Scientific, Waltham, USA) to prevent DNA contamination. Oligo(dT)₁₈ primers were used for cDNA synthesis. Customized polycistronic primers (Supplementary Dataset 11) were designed for polymerase chain reaction (PCR). AccuPower® PCR PreMix cat. #K-2012 (Bioneer, Daejeon, Korea) was used for PCR and PCR products were purified with LaboPass™ PCR Purification Kit cat. #CMR0112 (Cosmo Genetech, Seoul, Korea) for Sanger sequencing (Macrogen, Seoul, Korea).

## Reporting summary

Further information on research design is available in the Nature Portfolio Reporting Summary linked to this article.

# Data availability

The sequencing data generated in this study are deposited in the NCBI Sequence Read Archive under BioProject PRJNA851236 (https://www.

ncbi.nlm.nih.gov/bioproject/PRJNA851236). The complete genome of each species is available at the NCBI GenBank under the accession numbers listed below; JANCYW000000000 (*Cyanidium caldarium* [https://www.ncbi.nlm.nih.gov/nuccore/JANCYW000000000]), JAN-CYV000000000 (*Cyanidiococcus yangmingshanensis* [https://www.ncbi.nlm.nih.gov/nuccore/JANCYV000000000]), and JAN-CYU000000000 (*Galdieria sulphuraria* [https://www.ncbi.nlm.nih.gov/nuccore/JANCYU000000000]). Source Data used to generate all main text and supplementary figures can also be found in the Dryad dataset https://doi.org/10.5061/dryad.cfxpnvx7b. Source data are provided with this paper.

## Code availability
The codes used in this study are available in the Dryad repository https://doi.org/10.5061/dryad.cfxpnvx7b.

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

## Acknowledgements

This study was supported by a research grant from the National Research Foundation of Korea (NRF-2017R1A2B3001923, NRF-2022R1A2B5B03002312, and NRF-2022R1A5A1031361 to H.S.Y.). D.B. is supported by the National Aeronautics and Space Administration (NASA; 80NSSC19K0462) and a National Institute of Food and Agriculture-US Department of Agriculture Hatch grant (NJ01180). We gratefully acknowledge Kwi Young Han (GEOMAR Helmholtz Centre for Ocean Research Kiel), Hyun Ju Jung (Yonsei University), Sooyeon Park (Yonsei University), Seokwan Choi (Sungkyunkwan University), Louis Graf (IBENS; Institut de Biologie de l'École Normale Supérieure) and Eduard Ocaña-Pallarès (Eötvös Loránd University) for their comments and discussions, as well as Dongseok Kim (Sungkyunkwan University) for his assistance with the red algal photo.

## Author contributions

C.H.C., S.I.P., and H.S.Y. designed research; C.H.C., S.I.P., Y.L., T.-Y.H., and C.C. provided cultured samples for the experiment; C.H.C., S.I.P., Y.L., T.-Y.H., and H.C.Y., performed experiments; C.H.C., S.I.P., Y.L. analyzed data; and C.H.C., S.W.Y., D.B., and H.S.Y. wrote the manuscript. All authors have read and edited the final manuscript.

## Competing interests

The authors declare no competing interests.
