## [Peer Review File · Nature Communications]

Genome-wide signatures of adaptation to extreme environments in red algaeReviewers' Comments:

Reviewer #1:

Remarks to the Author:

In "Genome-wide signatures of adaptation to extreme environments", the authors present the sequencing, assembly, and annotation of three high-quality red alga nuclear genomes and gene model sets. The availability of such high-quality genomes and structural annotations, plus the included comparative genomic/transcriptomic analyses, are a significant addition to several genome- and alga-related fields. Since the chosen red algae are known for their extreme habitats, the authors equate species-specific genome alterations with adaptation to extreme lifestyles. This is a well-written manuscript that could be strengthened by avoiding the reader's impression that some results are anecdotal, as discussed below.

(1) There are multiple mentions of an 'extremophile toolkit' and inclusion of a cartoon to illustrate this toolkit, but it is not clear how the authors arrived at the gene set that comprises the toolkit or which genes are in the 'toolkit'. The data may be in one or more supplemental tables, but there needs to be quantification supporting such statements:

a. In the abstract: "We find that only a handful of genes in the 'extremophile toolkit' are shared by the two major orders, Cyanidiales and Galdieriales: i.e., most of the specialized genes evolved independently in each lineage." The authors should state the number of genes and provide a supplemental table showing which genes are in this toolkit. I assume the authors are referring to gene duplications in subtelomeric regions, but this could be made more explicit in the abstract. The authors should also quantify how many of those genes are conserved between the 3 species and how many are unique. This information doesn't have to go in the abstract, but there should be several sentences describing this gene set and a section in the Methods describing how this gene set is defined. An alternative would be to add small Venn diagrams with numbers in each part of Fig. 6 to show how many "HGT" from proks and the conservation between species, and how many genes resulted from STDG and the conservation between species. How the authors define conservation, i.e. OGFs as an example, should be included.

b. The authors should state how many genes in each species are predicted to have been acquired by HGT before and how many after the last common ancestor of Galdieriales and Cyanidiales. Based on Table S13, there are only 6 examples of HGT? Why is this number so much smaller than what was previously reported for *Galdieria sulphuraria* by Schönknecht, et al.; is it because of the methods used or differences between species?

(2) Throughout: the study would be strengthened by adding contextualization to findings. How do the observed differences between the genomes compare to non-extremophilic red algae and to other lineages? As an example, at line 125, "gene order was not strongly conserved between these red algal orders", is that statement unique to these orders or a general statement that can be said for other comparisons between orders of other algal lineages? The same applies to subtelomeric duplications and HGT; are the number of duplications / HGT events unique to these red algae/extremophiles or would a similar result with respect to number of subtelomeric duplications / HGT events be found with non-extremophile microbial eukaryotes? Another example is the statement referring to convergent evolution creating prokaryotic traits at line 413. These "prokaryotic" traits are not unique to these algae; for instance, using these criteria, some marine picoprasinophytes would be considered to be more prokaryotic-like (streamlined genomes, small number of genes, very few if any introns, very small cells). I would recommend removing and toning down these statements.

(3) How does the number of duplications in subtelomeric regions compare to the number of duplications in the rest of the chromosome? There are some places in the manuscript where the authors apply statistics to validate their observations (such as the length of intergenic regions), but such quantification and statistics are not applied throughout. As a result, many/most of the results sections come across as anecdotal.

(4) Is it coincidence that the major functional category of gene gain at branch 'a' and the major functional category of gene loss at branch 'b' was of 'O: posttranslational modification, protein turnover, chaperones'? Are the same OGFs contributing to these categories or different OGFs?

(5) Deposition: It wasn't immediately clear if the genome assembly and structural annotations, including predicted transcript and protein sequences will be made publicly available through Dryad. This data (which is the bulk of the work and a result of the authors expertise and diligence) should be made publicly (and easily) accessible.

(6) Polycistronic: the authors refer to Table S11, and I was expecting to find a list of identified proteins encoded by polycistronic transcripts in each of the three algal genomes, but the table is instead a list of primer sequences. There should be clarification or move the reference to Table S11 to the Supplemental note. If the authors are not providing a list of identified proteins encoded by polycistronic transcripts, can interested readers (who are not necessarily experts with Illumina and PacBio data) easily identify these from deposited data?

(7) Line 133: how many orthologs are conserved between species and how many paralogs are predicted?

Reviewer #2:

Remarks to the Author:

This submitted paper describes three new T2T genomes of cyanidiophycean red algae, of which habitats are polyextreme environments such as hot springs. The authors analyzed the genomic features and found subtelomeric duplications resulting in duplications of resided genes. Some of the duplicated subtelomeric genes are those for environmental adaptations such as heavy metal tolerance. They also performed comparative genomics of gene contents in Archaeplastida, and detected drastic gene losses in the last common ancestor of Cyanidiophyceae and further differential gene losses and gene gains in two lineages of Cyanidiophyceae. Thus, regardless of the same habitat, the two cyanidiophycean lineages possess different gene sets and exhibit different tolerance against environmental stresses, the latter of which was revealed by cultivation experiments. Consequently, this paper illuminates the "power of local selection" for eukaryotic genome evolution. The above findings are novel and of general interest of the readers. This research will significantly affect to future genome researches of eukaryotic algae and protists that occupy most of the diversity in the eukaryotic tree of life. I do not have any strong objection against most of the analyses and implications in this paper. But I would like to suggest to tone down or to conduct some additional experiments or analyses in certain points.

-Frequency of gene duplications

Although HGT and duplication of genes for heavy metal tolerance have been reported in several algae (e.g., Hirooka et al. 2017 PNAS E8304-E8313), subtelomeric gene duplication contributing to evolution of environmental stress tolerance is one of the novel findings in this paper. Thus, subtelomeric duplications should have contributed to evolutionary adaptation of the extremophile algae to hot spring conditions. But I would like to see more data to focus on how relatively important the subtelomeric duplications are for genome evolution of them. Duplications of any genes could also be counted from non-telomeric regions. I expect that there are few duplications in non-telomeric and non-sub-telomeric regions of the red algal chromosomes as their genomes are highly reduced. This might help quantification of importance of gene duplication in the sub-telomeric regions, further adding novel findings not only from functional importance of sub-telomeric gene duplications but also from frequencies of gene duplication events in genome evolution of the red algae.

-Selections of duplicated genes

Purifying selection in speciation but positive or relaxed selection in duplication for *merA* genes is interesting. But I would like to know whether this is the only exception that the authors could have detected the selection by using Ka/Ks ratios or this is one example of genes under selection detectable in the analysis. If the latter is the case, more examples should be shown. The archaeal-derived ATPase

genes in *Galdieria* would be interesting if they are under positive or purifying selections by analyzing Ka/Ks among the subtelomeric duplicated ATPase genes. If the former is the case, it would be better to clearly mention so.

Relevant to this, *merA* should be spelled out in line 234 as this is the first emergence of the gene.

-Adaptative evolution of proteins to thermophilic conditions

The authors detected the different types of heat shock protein genes from the red algal genomes. They found the number of chaperon genes are almost same among the genomes. However, expression level of the chaperon genes would be more directly involved in thermophilic lifestyles; the chaperon genes might be more expressed (or transcribed) in Cyanidiophyceae than mesophilic red algae. This is the case in some extremophile green algae (e.g., Hirooka et al. 2017 PNAS E8304-E8313). As the authors have already had RNAseq data of the thermophilic red algae and some mesophilic red algal RNAseq data are publicly available, it would provide more insight into contribution or no contribution of chaperons to thermophilic lifestyles.

Relevant to this, I have a concern about the *in silico* analysis of protein aggregation. Aggregation and folding of proteins could be affected by pH and ionic strength in addition to temperature. Indeed, I found TANGO v.2.3.1 has options to set pH and ionic strength. There is no mention about the settings and how the authors know the intracellular pH and ionic strength in cells of Cyanidiophyceae. These factors might be different between extremophiles and mesophilic species.

In addition, it would be better to show genomic evidence of specific adaptation in proteostasis machineries in the cyanidiophycan species as discussed in the supplementary note.

Accordingly, the current analyses and interpretation of the results seem insufficient to conclude anything about extremophilic adaptation of proteins. I would suggest delete this paragraph from the manuscript as it is not directly relevant to the main topic of the research which is introduced in the well-organized Abstract.

-Prokaryotic features

I agree with the authors that the thermophilic red algal genomes have acquired the reduced genomes, HGTs contributing to the adaptive evolution, small numbers of introns and spliceosomal components, lack of miRNA processing, and polycistronic expression of some genes. Some of them could allow the algae to thrive in hot springs. But it may remain unclear whether all these indeed contribute to the adaptative evolution to hot spring environments. These traits can be seen in other eukaryotes that do not thrive in hot springs, and thus, acquisition of one or more prokaryotic traits might be irrelevant to thriving in hot springs. Some parasitic protists such as *Giardia* possess reduced genomes with few introns and reduced sets of spliceosome components (Morrison et al. 2007 Science 317:1921-1926). HGTs for adaptation to certain environments, lack of miRNA processing, and polycistronic expression have been reported for other eukaryotes in previously published papers as cited in this submitted paper. I agree the above traits might be acquired by certain environmental pressures in extreme conditions but not limited or specific to hot springs. I would like to suggest to tone down in this paragraph or delete this paragraph from the manuscript as it is not directly relevant to the main topic of the research which is introduced in the well-organized Abstract.

In addition, I have a comment on the term "prokaryote-like features." As either of these traits is present in other eukaryotes, it is difficult to agree that all the above features are categorized as "prokaryote-like features."

-Metabolic pathway maps in Figs. 4 and 5

These figures seem to be models from prokaryotic cells. I would like to see models of mercuric and arsenic detoxifications in the eukaryotic algae. The protein sequences for the pathways could be predicted their possible localization by analyzing N-terminal peptides and internal transmembrane regions. I understand it might be difficult to predict proper localizations of eukaryotic proteins as eukaryotes possess multiple organelles. But, model pathways and localizations could be proposed by the analyses as the metabolic flow is for detoxification in this case.

-Minor points

line 141

Size increase of the intergenic regions might have happened in Cyanidioschyzon if the average size of intergenic regions in Cyanidium is as small as that in Cyanidiococcus. It would be more persuasive by describing the corresponding feature in Cyanidium in the main text.

line 193

I cannot follow what the "lack of conservation of subtelomeric duplicated genes in the last common ancestor of Cyanidiales" exactly means. Does it mean that subtelomeric duplicated genes in some chromosomes are distinct from those in other chromosomes in the last common ancestor of Cyanidiales, followed by differential inheritance of subtelomeric duplicated genes into the two Cyanidiales lineages?

line 392

I cannot follow why certain evolutionary constraints result in gene loss. Constraints would function for gene retention or against gene loss. Is it proper to say "evolutionary pressure" in this case?

-Typos/Wording

line 52

Delete the period from "a. variety of."

line 219

"Linker protein" should be "linker peptide."

Lines 330 and 354

What does "host" mean in this context?

Line 381

Spell out "pri-miRNA" as this is the first emergence of this wording.

**Point-by-point response**

**Reviewer #1:**

In “Genome-wide signatures of adaptation to extreme environments”, the authors present the sequencing,
assembly, and annotation of three high-quality red alga nuclear genomes and gene model sets. The availability
of such high-quality genomes and structural annotations, plus the included comparative genomic/
transcriptomic analyses, are a significant addition to several genome- and alga-related fields. Since the chosen
red algae are known for their extreme habitats, the authors equate species-specific genome alterations with
adaptation to extreme lifestyles. This is a well-written manuscript that could be strengthened by avoiding the
reader's impression that some results are anecdotal, as discussed below.

**[Response]**

We thank the reviewer for these positive comments about our work, as well as for the detailed
recommendations for improvement of our manuscript. These suggestions have strengthened the ideas in our
manuscript. In response to the reviewer's comment, we updated the materials used in new analyses and
uploaded genome data to public database that is easily accessible.

# Reviewer link:

- • Dryad dataset (doi: 10.5061/dryad.cfxpnvx7b):

<https://datadryad.org/stash/share/wG12T-8kTGCxrTffWD-vTVbRL2IDPzkIUfA2tUQTJus>

- • NCBI (BioProject: PRJNA851236; release data: Aug 31, 2022)

<https://dataview.ncbi.nlm.nih.gov/object/PRJNA851236?reviewer=8a2tavah84stp8hvirm6ts0kq6>

(1) There are multiple mentions of an ‘extremophile toolkit’ and inclusion of a cartoon to illustrate this toolkit,
but it is not clear how the authors arrived at the gene set that comprises the toolkit or which genes are in the
‘toolkit’. The data may be in one or more supplemental tables, but there needs to be quantification supporting
such statements:

a. In the abstract: “We find that only a handful of genes in the ‘extremophile toolkit’ are shared by the two
major orders, Cyanidiales and Galdieriales: i.e., most of the specialized genes evolved independently in each
lineage.” The authors should state the number of genes and provide a supplemental table showing which
genes are in this toolkit. I assume the authors are referring to gene duplications in subtelomeric regions, but
this could be made more explicit in the abstract. The authors should also quantify how many of those genes
are conserved between the 3 species and how many are unique. This information doesn’t have to go in the
abstract, but there should be several sentences describing this gene set and a section in the Methods describing
how this gene set is defined. An alternative would be to add small Venn diagrams with numbers in each part
of Fig. 6 to show how many “HGT” from proks and the conservation between species, and how many genes
resulted from STDG and the conservation between species. How the authors define conservation, i.e. OGfs as
an example, should be included.

**[Response]**

We meant the term ‘extremophile toolkit’ to refer to a collection of adaptation strategies for extreme
environments, rather than a collection of specific genes. We discovered that the term ‘extremophile toolkit’
can be misconstrued due to its terminology ('toolkit') and some of our previous statements, including the
sentence mentioned by the reviewer. To minimize the misunderstanding, we replace it as “environment
adaptation strategies”.

We provided most of the details (e.g., orthogroups and genes) for subtelomeric gene duplication (STGD)
in supplementary tables (S5-S7), however, as suggested by the reviewer, we have also updated information in
figures (e.g., Figure 2a: Venn diagram of STGD orthogroup), tables, and supplementary data (Dryad database:
materials used for analysis). More information about our study can be found in other comments (see below).
We made the following changes to the sentences and figures,

- • Page 2, Line 18-19: “These extremophilic adaptation strategies are shared by the two major orders, Cyanidiales and Galdieriales, but most of the specialized genes evolved independently in each lineage.”
- • Page 2, Line 21-23: “... and demonstrate that the genomic consequence of extremophilic adaptation varies among the taxa in different microhabitats.”
- • Page 9-10, Line 196-201: “The STGD ratio was calculated to determine its impact on recent gene duplication events (Supplementary Fig. 10). STGD accounted for 28.9-31.9% of recent gene duplications in both Cyanidiales and Galdieriales, and Fisher's exact test (p -value 0.05) supported the correlation between gene duplication and subtelomeric region. As a result, recent gene duplications of Cyanidiophyceae species have been significantly influenced by STGD events.”
- • New figure added: added **Figure 2a**
- • New figure added: **Supplementary Figure 8**
- • Page 26-27, Line 610-624: “**Identification of subtelomere and gene duplication ratio** To identify subtelomeric regions from genomes, LASTZ alignment v7.0.2 were used to see if there were any conserved regions between chromosomes ¹⁰³. Subtelomere regions near telomeric repeats were manually confirmed using LASTZ alignments across chromosomes, and subtelomeric genes were identified from subtelomeric regions (Supplementary Fig. 8). We attempted to remove paralogs from gene duplication detection and focus on recent gene duplications in order to calculate the proportion of subtelomeric gene duplication when compared to the overall number of gene duplication. DIAMOND v2.0.5.143 with variable parameters was applied to conduct protein homology searches (blastp) between each protein sequence in the entire proteomes. Query and subject coverage (‘-q’, ‘-s’) were set to 70-90% with 5% intervals, and protein identity (‘-i’) was set to 70-90% with 5% intervals as well. As a result, this analysis used a total of 25 parameter combinations, which were visualized in a plot (Supplementary Fig. 10; Supplementary Table 13). Fisher's exact test (‘fisher.test’), implemented in R, was used to independently test the significance of subtelomeric regions and gene duplication in each species.”
- • New table added: **Supplementary Table 13**

b. The authors should state how many genes in each species are predicted to have been acquired by HGT before and how many after the last common ancestor of Galdieriales and Cyanidiales. Based on Table S13, there are only 6 examples of HGT? Why is this number so much smaller than what was previously reported for *Galdieria sulphuraria* by Schönknecht, et al.; is it because of the methods used or differences between species?

**[Response]**

Table S13 (previous) included the information about the genes identified in this study. Regarding the examination of horizontal gene transfer (HGT), we do not delve deeply into this topic in this paper, due to: i) HGTs have already been comprehensively studied in previous papers (Schönknecht *et al.*, *Science*, 2013; Rossoni *et al.*, *eLife*, 2019; Van Etten *et al.*, *Trends Genet*, 2020), ii) Some phylogenies are poorly supported as HGTs (e.g., low bootstrap support values, long branches). As a result, we focused only on the genes that needed to be discussed in relation to their known functions. The Dollo parsimony result (Figure 3) using orthogroups from representative Archaeplastida indirectly illustrates the number of HGT between lineages. However, as previously stated, we found that determining the precise number of HGTs was difficult due to methodological issues, so we chose to focus on genes that have been functionally identified and have a clear phylogenetic signal of foreign origin. We revised the sentence as follows,

- • Page 2, Line 18-19: “These extremophilic adaptation strategies are shared by the two major orders, Cyanidiales and Galdieriales, but most of the specialized genes evolved independently in each lineage.”
- • Page 2, Line 21-23: “... and demonstrate that the genomic consequence of extremophilic adaptation varies among the taxa in different microhabitats.”

- • Removed previous Table S13 from Supplementary Table: Dryad database still contains orthogroup
information.

(2) Throughout: the study would be strengthened by adding contextualization to findings. How do the
observed differences between the genomes compare to non-extremophilic red algae and to other lineages? As
an example, at line 125, “gene order was not strongly conserved between these red algal orders”, is that
statement unique to these orders or a general statement that can be said for other comparisons between orders
of other algal lineages? The same applies to subtelomeric duplications and HGT; are the number of
duplications / HGT events unique to these red algae/extremophiles or would a similar result with respect to
number of subtelomeric duplications / HGT events be found with non-extremophile microbial eukaryotes?

**[Response]**

Extremophilic red algae (0.73-0.80%) do not have a markedly higher number of HGT-derived genes when
compared to other lineages (e.g., mesophilic red algae: 1.72-6.49%), according to a previous report on HGTs
in various eukaryotic lineages (Van Etten *et al.*, *Trends Genet*, 2020). However, our main thrust is that
because extremophilic red algae have undergone genome reduction (a process that reduces the genome size
and number of genes), they should have lost the majority of the genes that were not required for the host.
Despite the massive gene loss, they continue to have gene duplications and acquire new functions, which is
therefore likely to be essential for survival in hostile environments.

To compare STGD among extremophilic and non-extremophilic red algae, it is necessary to have
chromosomal level genome data. This is lacking for other red algae and most genomes are fragmented (e.g.,
*Chondrus crispus*, *Porphyra umbilicalis*, *Chondria armata*) or gene model information is unavailable (e.g.,
*Neoporphyra haitanensis*), excepting *Porphyridium purpureum* genome (Lee *et al.*, *Nat Commun*, 2019).
Because the genome of *Porphyridium purpureum* does not appear to have a clear subtelomeric region,
comparative analysis was difficult in this case. This is also applicable when comparing gene order across
different lineages (i.e., between orders). If highly resolved (e.g., chromosomal level) assemblies of mesophilic
red algae genomes are released in the future, we would like to conduct such comparisons. Therefore, we did
not examine the entire set of HGTs in this paper.

Another example is the statement referring to convergent evolution creating prokaryotic traits at line 413.
These “prokaryotic” traits are not unique to these algae; for instance, using these criteria, some marine
picoprasinophytes would be considered to be more prokaryotic-like (streamlined genomes, small number of
genes, very few if any introns, very small cells). I would recommend removing and toning down these
statements.

**[Response]**

We agree with both reviewer comments (“prokaryote-like features”). Based on these comments, we removed
or modified this section.

- • Page 19, Line 440-443: “Another unique trait present in some Cyanidiophyceae is expansion of
polycistronic ... in Dryad Supplementary Dataset.”

(3) How does the number of duplications in subtelomeric regions compare to the number of duplications in
the rest of the chromosome? There are some places in the manuscript where the authors apply statistics to
validate their observations (such as the length of intergenic regions), but such quantification and statistics are
not applied throughout. As a result, many/most of the results sections come across as anecdotal.

**[Response]**

This is a good point raised and we appreciate the opportunity to address the issue. To make it more clear about
subtelomeric gene duplication versus gene duplication, we performed the analysis shown below. The STGD
ratio from the entire gene duplication is now shown. To exclude paralogs from the analysis, we used coverage
(70-90%) and identity (70-90%) on a scale with 25 different parameter combinations.

[Table] Example of dataset table used for a statistical test (Fisher's exact test) to determine the significance of subtelomeric regions in Cyanidiophyceae related to gene duplications. The average in this table means the average of values from 25 different combination parameters used for DIAMOND.

(each species)	Duplicated genes	Genes that are not duplicated
Subtelomeric region	(Average of STGDs)	(Subtelomere genes) - (Average of STGDs)
Non-subtelomeric region	(Average of duplicated genes) - (Average of STGDs)	(Total genes) - (Average of duplicated genes) - (Subtelomere genes) + 2 * (Average of STGDs)

Percentages of subtelomeric gene duplications (STGDs) versus gene duplications

DIAMOND (BLASTp): 25 different parameter combinations were used
- coverage (70, 75, 80, 85, 90%)
- identity (70, 75, 80, 85, 90%)

[Figure] The proportion of subtelomeric gene duplication when compared to overall gene duplication in four cyanidiophycean species. A total of 25 combinations of query & subject coverage and protein identity parameters were considered.

Since STGD accounted for ca. 30% of recent gene duplications, we infer that gene duplication in Cyanidiophyceae species has been highly influenced by STGD, which is supported by Fisher's exact test (p -value < 0.05). We added and revised the sentence as follows,

- Page 9-10, Line 196-201: "The STGD ratio was calculated to determine its impact on recent gene duplication events (Supplementary Fig. 10). STGD accounted for 28.9-31.9% of recent gene duplications in both Cyanidiales and Galdieriales, and Fisher's exact test (p -value 0.05) supported the correlation between gene duplication and subtelomeric region. As a result, recent gene duplications of

160 Cyanidiophyceae species have been significantly influenced by STGD events.”

• New figure added: **Supplementary Figure 8**

• Page 26-27, Line 610-624: “**Identification of subtelomere and gene duplication ratio** To identify
subtelomeric regions from genomes, LASTZ alignment v7.0.2 were used to see if there were any
conserved regions between chromosomes ¹⁰³. Subtelomere regions near telomeric repeats were
manually confirmed using LASTZ alignments across chromosomes, and subtelomeric genes were
identified from subtelomeric regions (Supplementary Fig. 8). We attempted to remove paralogs from
gene duplication detection and focus on recent gene duplications in order to calculate the proportion of
subtelomeric gene duplication when compared to the overall number of gene duplication. DIAMOND
v2.0.5.143 with variable parameters was applied to conduct protein homology searches (blastp) between
each protein sequence in the entire proteomes. Query and subject coverage (‘-q’, ‘-s’) were set to 70 to
90% with 5% intervals, and protein identity (‘-i’) was set to 70-90% with 5% intervals as well. As a
result, this analysis used a total of 25 parameter combinations, which were visualized in a plot
(Supplementary Fig. 10; Supplementary Table 13). Fisher's exact test ('fisher.test'), implemented in R
was used independently to test the significance of subtelomeric regions and gene duplication in each
species.”

• New table added: **Supplementary Table 13**

(4) Is it coincidence that the major functional category of gene gain at branch ‘a’ and the major functional
category of gene loss at branch ‘b’ was of ‘O: posttranslational modification, protein turnover, chaperones’?
Are the same OGFs contributing to these categories or different OGFs?

**[Response]**

To clarify functional categories, we updated the detailed orthogroup, GO (Gene Ontology), and COG
(Clusters of Orthologous Genes) information for each gain and loss event in key lineages (Fig. 3, see files in
‘[2-Figure] Fig. 3, S12 - DolloP, COG/Fig. S12 - GO, COG, egglog’ directory in Dryad data).

# List of orthogroups and their functional annotations: “[branch info.]_[gain or loss]_egglog_annot.txt” files
(‘...’: or the sake of feasibility, some information has been omitted.)

# Orthogroup	GO	COG	Description
OG0000461	-,...,GO:0044464	C,J,O	"protein-chromophore linkage", ..., "PFAM Phycobilisome protein"
OG0000708	GO:0003002,...,GO:1901564	M	"no hit", "intein-mediated protein splicing"
...			

# Gene ontology information of orthogroups: “[branch info.]_[gain or loss]_GO_count.txt” files

# GO	COUNT	DEFINITION
GO:0005575	164	cellular_component
GO:0005622	158	intracellular anatomical structure
...		

COGs are classified into 37 categories, therefore this information is overly broad and appears to be a
coincidence of functional categories in various branches. This is also similarly applied for GOs, with top
hierarchy GOs (those with broader functions) being counted more than bottom hierarchy GOs (more specific
functions). We did not go into detail about functional categories based on this information, instead focusing
on specific orthogroups or genes whose function has been clearly identified in previous studies. We have now
included a link in this response to the reviewer's request for the Dryad data.

# Reviewer link:

• <https://datadryad.org/stash/share/wG12T-8kTGCxrTffWD-vTVbRL2IDPpkIUfA2tUQTJus>

(5) Deposition: It wasn't immediately clear if the genome assembly and structural annotations, including
predicted transcript and protein sequences will be made publicly available through Dryad. This data (which is
the bulk of the work and a result of the authors expertise and diligence) should be made publicly (and easily)
accessible.

**[Response]**

As suggested by the reviewer, we have uploaded our raw data and genomes to the NCBI database (BioProject:
PRJNA851236).

- • *Cyanidium* [SAMN29217976]: SRR19760114, SRR19760115, SRR19760121, SRR19760122,
JANCYW000000000
- • *Cyanidiococcus* [SAMN29217977]: SRR19760116-SRR19760119, JANCYV000000000
- • *Galdieria* [SAMN29217978]: SRR19760108-SRR19760113, SRR19760120, JANCYU000000000

# Reviewer link:

- • <https://dataview.ncbi.nlm.nih.gov/object/PRJNA851236?reviewer=8a2tavah84stp8hvirm6ts0kq6>

NCBI submission for gene annotation and genomic features requires a specialized format, which has been
delayed thus far. The preparation of remaining materials for the NCBI submission is an ongoing process, and
we will manage this issue during the revision period. Based on this information, we updated '**Data**
**availability**' section in the manuscript as follows,

- • Page 29, Line 676-686: "... BioProject PRJNA851236. The complete genomes of each species were
assigned the NCBI GenBank accession numbers listed below; *Cyanidium caldarium*
(JANCYW000000000), *Cyanidiococcus yangmingshanensis* (JANCYV000000000), and *Galdieria*
*sulphuraria* (JANCYU000000000). All of the other data used and generated in this study including
genome information are deposited in the Dryad database (<https://doi.org/10.5061/dryad.cfxpvnv7b>). ..."

(6) Polycistronic: the authors refer to Table S11, and I was expecting to find a list of identified proteins
encoded by polycistronic transcripts in each of the three algal genomes, but the table is instead a list of primer
sequences. There should be clarification or move the reference to Table S11 to the Supplemental note. If the
authors are not providing a list of identified proteins encoded by polycistronic transcripts, can interested
readers (who are not necessarily experts with Illumina and PacBio data) easily identify these from deposited
data?

**[Response]**

We agree with the reviewer that a list of identified proteins encoded by polycistronic transcripts might be of
interest to readers. Files included with Dryad data contain a list of those identified reads and the encoded
proteins they correspond to (see '.fasta' and '.tbl' files in '[2-Figure] Fig. S19 -
Polycistronic/Polycistronic_identification' directory in Dryad data).

# Identified polycistronic transcripts and functional gene category (COG) of those genes:

"PCT_CCYA_CDCA.xlsx" file

# Transcript ID	Length	Gene	COG Category
P4_CCYA01_LOC_166099-170502_F	4403	CCYA01G0065	G
P4_CCYA01_LOC_166099-170502_F	4403	CCYA01G0064	.
...			

# Identified polycistronic transcripts and encoded genes: "[species]_polycistronic_transcript_list.tbl" files

# Iso-Seq read	Mapped length	Number of genes	Gene list
----------------	---------------	-----------------	-----------

P4_CCYA01_LOC_166099-170502_F	4403	2	CCYA01G0065,CCYA01G0064
P4_CCYA01_LOC_184575-188936_R	4361	2	CCYA01G0072,CCYA01G0071
...			

We revised the sentence as follows,

- • Page 19, Line 442-443: "... (Supplementary Note 8; Supplementary Fig. 20; Supplementary Table 11;
 a list of identified proteins encoded by polycistronic transcripts are provided in Dryad Supplementary
 Dataset)."

(7) Line 133: how many are orthologs are conserved between species and how many paralogs are predicted?

**[Response]**

Based on the criteria used, the ortholog and paralog number can vary. Based on the orthogroup result, we
 identified a total 4,102/4,608 (89.0%) orthogroups that were shared in all three species. From these results,
 paralogs (species-specific orthogroups and unassigned genes) were detected 242/4,832 (5.01%) genes in
 *Cyanidiococcus*, 388/4,870 (7.97%) genes in *Cyanidium*, and 191/4,757 (4.02%) genes in *Cyanidioschyzon*.
 However, as previously stated, the number of paralogs and orthologs cannot be clearly described without
 careful examination of individual phylogenetic trees, and sampling of more Cyanidiales species may change
 these results. Therefore, we are unable to clearly define orthologs and paralogs based on our analysis and have
 decided to not present these numbers in the revised manuscript.

**Reviewer #2:**

This submitted paper describes three new T2T genomes of cyanidiophycean red algae, of which habitats are
 polyextreme environments such as hot springs. The authors analyzed the genomic features and found
 subtelomeric duplications resulting in duplications of resided genes. Some of the duplicated subtelomeric
 genes are those for environmental adaptations such as heavy metal tolerance. They also performed
 comparative genomics of gene contents in Archaeplastida, and detected drastic gene losses in the last
 common ancestor of Cyanidiophyceae and further differential gene losses and gene gains in two lineages of
 Cyanidiophyceae. Thus, regardless of the same habitat, the two cyanidiophycean lineages possess different
 gene sets and exhibit different tolerance against environmental stresses, the latter of which was revealed by
 cultivation experiments. Consequently, this paper illuminates the "power of local selection" for eukaryotic
 genome evolution. The above findings are novel and of general interest of the readers. This research will
 significantly affect to future genome researches of eukaryotic algae and protists that occupy most of the
 diversity in the eukaryotic tree of life. I do not have any strong objection against most of the analyses and
 implications in this paper. But I would like to suggest to tone down or to conduct some additional experiments
 or analyses in certain points.

**[Response]**

We would like to thank the reviewer for their interest in our work, as well as for the detailed comments and
 in-depth review. The reviewer's comments have improved our manuscript. We uploaded data and materials
 used in the updated manuscript to the Dryad database (see the link below).

# Reviewer link:

- • Dryad dataset (doi: 10.5061/dryad.cfxpnvx7b):
 <https://datadryad.org/stash/share/wG12T-8kTGCxrTffWD-vTVbRL2IDPxxkIUfA2tUQTJus>

-Frequency of gene duplications

Although HGT and duplication of genes for heavy metal tolerance have been reported in several algae (e.g.,
 Hirooka et al. 2017 PNAS E8304-E8313), subtelomeric gene duplication contributing to evolution of

environmental stress tolerance is one of the novel findings in this paper. Thus, subtelomeric duplications
should have contributed to evolutionary adaptation of the extremophile algae to hot spring conditions. But I
would like to see more data to focus on how relatively important the subtelomeric duplications are for genome
evolution of them. Duplications of any genes could also be counted from non-telomeric regions. I expect that
there are few duplications in non-telomeric and non-sub-telomeric regions of the red algal chromosomes as
their genomes are highly reduced. This might help quantification of importance of gene duplication in the sub-
telomeric regions, further adding novel findings not only from functional importance of sub-telomeric gene
duplications but also from frequencies of gene duplication events in genome evolution of the red algae.

**[Response]**

Please see the response above to question (3) of reviewer 1.

-Selections of duplicated genes

Purifying selection in speciation but positive or relaxed selection in duplication for merA genes is interesting.
But I would like to know whether this is the only exception that the authors could have detected the selection
by using Ka/Ks ratios or this is one example of genes under selection detectable in the analysis. If the latter is
the case, more examples should be shown. The archaeal-derived ATPase genes in *Galdieria* would be
interesting if they are under positive or purifying selections by analyzing Ka/Ks among the subtelomeric
duplicated ATPase genes.

If the former is the case, it would be better to clearly mention so.

Relevant to this, merA should be spelled out in line 234 as this is the first emergence of the gene.

**[Response]**

We listed 21 pair of candidate genes in Supplementary Table 7, however, gene pairs in the STGD did not
show evidence of purifying selection (not supported by Fisher test), whereas non-STGD pairs show a
significant signal of purifying selection. Due to this difference, we were unsure how to interpret these results.

Meanwhile, for archaeal-derived ATPase genes, some issues (e.g., unresolved relationship of archaeal
and *Galdieria* ATPase genes) have been arisen in a previous paper (Rossoni *et al.*, *eLife*, 2018), therefore, we
have only mentioned their subtelomeric gene duplication. We also reconstructed phylogenies of archaeal-
derived ATPase genes for our analysis, but we were unable to provide a clear answer due to low bootstrap
values, phylogenetic inconsistency, and unclear taxon sampling (no significant archaeal hits were retrieved).
Therefore, we chose only merA as a candidate for this analysis because, i) we focused on genes whose
functions we already knew, and ii) the phylogeny of genes was clear.

-Adaptative evolution of proteins to thermophilic conditions

The authors detected the different types of heat shock protein genes from the red algal genomes. They found
the number of chaperon genes are almost same among the genomes. However, expression level of the
chaperon genes would be more directly involved in thermophilic lifestyles; the chaperon genes might be more
expressed (or transcribed) in Cyanidiophyceae than mesophilic red algae. This is the case in some
extremophile green algae (e.g., Hirooka *et al.* 2017 PNAS E8304-E8313). As the authors have already had
RNAseq data of the thermophilic red algae and some mesophilic red algal RNAseq data are publicly
available, it would provide more insight into contribution or no contribution of chaperons to thermophilic
lifestyles.

**[Response]**

We agree that chaperones play a role in adaptation to harsh environments, however, our genome data
(included in supplementary information) could not be used to address this issue. Based on this reviewer's
comments, we collected data on mesophilic red algae (e.g., *Gracilariopsis chorda*, *Porphyridium purpureum*)
and calculated the expression levels for each species under control conditions. For example, we have provided

a table (see below) of one chaperone protein family (DnaJ) that exists along with a calculation of their
expression level (TPM values provided in the Dryad database).

# Expression level of the chaperone protein coding gene, *DnaJ* homolog (same orthogroup; OG0002259) in
control condition of each species.

Speices + Strain	CDS Name	Length	Effective Length	TPM	NumReads
Galdieria sulphuraria 108.79 E11	GASU_P_CTG32_G0837	1482	1312.401	11.577502	474.551
Galdieria sulphuraria 108.79 E11	GASU_SCF00_G1823	1482	1312.401	18.584542	761.764
Cyanidiococcus yangmingshanensis 8.1.23 F7	CCYA04G1202	1812	1634.083	16.772039	281.015
Cyanidium caldarium 063 E5	CDCA15G4056	1614	1454.044	101.625029	10796.011
Gracilariopsis chorda	GRC0021GENE2390	1557	1388.32	23.4283	659
Porphyridium purpureum CCMP1328	POR1872...scf295_9	1740	1490	1.925653	18

However, we are unable to find any correlation (expected to have higher TPM values in extremophilic
red algae) and therefore do not think this analysis provides useful information to readers: i.e., the expression-
level comparison among species should be compared to uniform control conditions, however, the above
comparison was made from independent experiments with different conditions. Furthermore, our RNA-seq
data were primarily collected to allow gene model validation. The above table is only for the reviewer's
reference and is not included in the main text. However, we will take this comment into account for
upcoming analyses.

Relevant to this, I have a concern about the in silico analysis of protein aggregation. Aggregation and folding
of proteins could be affected by pH and ionic strength in addition to temperature. Indeed, I found TANGO
v.2.3.1 has options to set pH and ionic strength. There is no mention about the settings and how the authors
know the intracellular pH and ionic strength in cells of Cyanidiophyceae. These factors might be different
between extremophiles and mesophilic species.

**[Response]**

As this reviewer notes, pH and ionic strength may be important parameters when using TANGO. We used
three different values for the temperature parameter (290K, 300K, and 320K), but we used the default option
for the pH 7 and ionic strength 0.1 parameters. The reason for using the default parameter for pH is that we
already know the internal pH of the extremophile *Cyanidium caldarium* (pH 6.64 ± 0.09) and that when
compared to other mesophilic algae (pH 5.0-7.4), they do not show a significantly lower pH (Beardall &
Entwisle, *Phycologia*, 1984). We were not able to obtain ion strength information except from a study about
*Cyanidium caldarium* proteins in which they used an ionic strength of 1.0 by adding NaCl, but this was not
about the internal condition of cells (Enami, *Plant Cell Physiol*, 1978). Because all parameters will be applied
uniformly to all Archaeplastida proteomes for the aggregation prone propensity *in silico* survey, we
respectfully did not consider it critical for our comparative analysis. In addition, the paper from which we got
the majority of our ideas for this analysis (Draceni & Pechmann, *PNAS*, 2019) used default parameters.
As this reviewer suggests, we also tested with pH 6.6 (intracellular pH referred to in the previous article),
300K temperature, and 0.1 ionic strength.

[Figure] Estimated value of aggregation-prone regions using proteomes of Archaeplastida (248,192 proteins).

The intracellular pH of *Cyanidium caldarium* (pH 6.6) was used as the pH parameter, and the other

parameters were set to the default.

The new findings also revealed consistent differences between mesophilic and extremophilic red algae (t-test: $df=36,536$, $p\text{-value} < 0.05$) (see above figure only for this reviewer's reference). Given all of the information presented above, our analysis is applicable. We updated and revised the sentence as follows,

- SI Page 11, line 260-263: “When compared to other mesophilic algae (pH 5.0-7.4), the internal pH of the extremophile *Cyanidium caldarium* (pH 6.64 ± 0.09) do not show any significantly lower pH⁴⁴. Because there was no ion strength measurement, we used the default pH and ionic strength parameters.”

In addition, it would be better to show genomic evidence of specific adaptation in proteostasis machineries in the cyanidiphycean species as discussed in the supplementary note. Accordingly, the current analyses and interpretation of the results seem insufficient to conclude anything about extremophilic adaptation of proteins. I would suggest delete this paragraph from the manuscript as it is not directly relevant to the main topic of the research which is introduced in the well-organized Abstract.

[Response]

We agree with the reviewer's comment. Based on the comment, we deleted this paragraph.

-Prokaryotic features

I agree with the authors that the thermophilic red algal genomes have acquired the reduced genomes, HGTs contributing to the adaptive evolution, small numbers of introns and spliceosomal components, lack of miRNA processing, and polycistronic expression of some genes. Some of them could allow the algae to thrive in hot springs. But it may remain unclear whether all these indeed contribute to the adaptive evolution to hot spring environments. These traits can be seen in other eukaryotes that do not thrive in hot springs, and thus, acquisition of one or more prokaryotic traits might be irrelevant to thriving in hot springs. Some parasitic protists such as *Giardia* possess reduced genomes with few introns and reduced sets of spliceosome components (Morrison et al. 2007 Science 317:1921-1926). HGTs for adaptation to certain environments, lack

of miRNA processing, and polycistronic expression have been reported for other eukaryotes in previously
published papers as cited in this submitted paper. I agree the above traits might be acquired by certain
environmental pressures in extreme conditions but not limited or specific to hot springs. I would like to
suggest to tone down in this paragraph or delete this paragraph from the manuscript as it is not directly
relevant to the main topic of the research which is introduced in the well-organized Abstract.

In addition, I have a comment on the term “prokaryote-like features.” As either of these traits is present in
other eukaryotes, it is difficult to agree that all the above features are categorized as “prokaryote-like
features.”

**[Response]**

We agree with the reviewer comment (“prokaryote-like features”). Based on reviewer comment, we removed
this sentence.

-Metabolic pathway maps in Figs. 4 and 5

These figures seem to be models from prokaryotic cells. I would like to see models of mercuric and arsenic
detoxifications in the eukaryotic algae. The protein sequences for the pathways could be predicted their
possible localization by analyzing N-terminal peptides and internal transmembrane regions. I understand it
might be difficult to predict proper localizations of eukaryotic proteins as eukaryotes possess multiple
organelles. But, model pathways and localizations could be proposed by the analyses as the metabolic flow is
for detoxification in this case.

**[Response]**

As suggested by the reviewer, we analyzed transit peptide prediction (TargetP v1.1: ‘Plant’ option) and
transmembrane regions (DeepTMHMM v1.0.1) of heavy metal detoxification genes (checked major genes)
from four representative cyanidiophycean species. In this analysis, we excluded highly duplicated genes and
multifunctional transporters (e.g., Pst).

- TargetP: there were no targeting peptides predicted for any of the proteins.

- DeepTMHMM (protein type: TM/GLOB)

o merA: GLOB

o AQP: 6-8 TMs

o arsA: GLOB (1 out of 12 proteins had TM: CDCA11G3147)

o arsB: 11-14 TMs

o arsC: GLOB

o arsJ: 12 TMs

o arsM: GLOB

As a result, our pathway model appears to fit predicted results, and we have included these results in
our Dryad dataset (see files in ‘[4-ETC] Ab initio localization’ directory in Dryad data). We reconstructed the
metabolic pathway depicted in Figures 4 and 5 based on several papers that also discuss the eukaryotic
detoxification system for heavy metals. The arsenic detoxification pathway depicted in Figure 5 was verified
by several molecular experiments in *Cyanidioschyzon* (Qiu *et al.*, *PNAS*, 2009), a Cyanidiales species, and we
reconstructed the detoxification pathway using results from these experiments as well as information from
other articles discussing other eukaryotic species. Therefore, all of the current evidence supports our model,
which appears to be robust. We revised the sentence as follows,

• Page 16, Line 360-362: “... study as well as a few other arsenic detoxification pathway studies⁵²⁻⁵⁵. We
inspected these genes in other Cyanidiophyceae species and used transit peptide prediction and
transmembrane region prediction to confirm their possible localization.”

• Page 26, Line 607-608: “TargetP v1.1 and DeepTMHMM v1.0.1 were used to predict transit peptides
and transmembrane domain regions in order to validate gene localization.”

-Minor points
line 141

Size increase of the intergenic regions might have happened in Cyanidioschyzon if the average size of intergenic regions in Cyanidium is as small as that in Cyanidiococcus. It would be more persuasive by describing the corresponding feature in Cyanidium in the main text.

[Response]

As suggested by the reviewer, we now added intergenic regions of *Cyanidium* in our revised manuscript. We present a boxplot of intergenic regions of Cyanidiophyceae (including *Galdieria*) for reviewer, but we think it is not necessary to include it in the current manuscript, because there are other factors related to genome size increase (e.g., intron and telomeric regions, see below).

[Figure] Box plot of intergenic region distances between genes (bp). CCYA: 929.6762 bp, CDCA: 319.7097 bp, CZME: 1889.814 bp, GASU: 470.851 bp (CCYA: *Cyanidiococcus*, CDCA: *Cyanidium*, CZME: *Cyanidioschyzon*, GASU: *Galdieria*)

Introns and telomeric regions were not considered in this analysis because we defined intergenic region as sequences between two genes. All statistical tests supported *Cyanidioschyzon* is significantly different from others (p -value < 0.05) and our following analysis of repeat comparison support this idea. We revised the sentence as follows,

- Page 7, line 137-140: “Using a statistical approach (student’s t-test: p -value < 0.05), we discovered that the average intergenic region of different Cyanidiales species (CCYA: 929.7 bp, CZME: 1,889.8 bp, CDCA: 319.7 bp) is significantly different. The size of intergenic regions between sister species increased due to repeat expansion in ...”

line 193

I cannot follow what the “lack of conservation of subtelomeric duplicated genes in the last common ancestor of Cyanidiales” exactly means. Does it mean that subtelomeric duplicated genes in some chromosomes are distinct from those in other chromosomes in the last common ancestor of Cyanidiales, followed by differential inheritance of subtelomeric duplicated genes into the two Cyanidiales lineages?

[Response]

Based on this comment, we revised the sentence as follow:

- • Page 9, line 193-196: “This result indicates either subtelomeric duplicated genes in some chromosomes
are distinct from those in other chromosomes in the last common ancestor of Cyanidiales, followed by
differential inheritance of subtelomeric duplicated genes into the two Cyanidiales lineages, or that
STGDs occurred post-divergence of this order.”

Furthermore, we've now added Figure 2a (Venn diagram of STGD orthogroups) to help illustrate this
issue. As mentioned above, only ‘kelch, trefoil, and hedgehog’ orthogroup were shared and others do not
show overlap between all Cyanidiales species. This also applies to the ancestor of Cyanidiophyceae, because,
with the exception of a single orthogroup (GTP binding protein) that has been amplified independently in each
lineage (shown in Supplementary Fig. S9), there is no overlap in STGD events between Cyanidiales and
Galdieriales.

Some may be concerned about a few STGD overlaps found between Cyanidiales lineages
(particularly CCYA and CZME in Fig. 2a), but this can be explained by genetic distance between species.
CCYA and CZME are genetically more closely related compared to other Cyanidiales or Galdieriales species
(i.e., CDCA), and genomic synteny has been found to be more conserved in CCYA & CZME than that of
other Cyanidiales. We provided additional phylogenetic trees (see below figures for rebuttal purpose), not
present in our current manuscript, to illustrate the clearer genetic distance between species.

[Figure - Left] Summary maximum likelihood tree of Cyanidiophyceae inferred from plastid-encoded *rbcL*
gene sequences (Figure 3 from Etten *et al.*, *Semin Cell Dev Biol*, 2022), [Figure - Right] BI (MrBayes) & ML
(RAxML, IQ-TREE) phylogenetic tree using the alignment of 130 plastid protein sequences from 14
Cyanidiophyceae species and six of outgroup species (Unpublished).

As a result of their evolutionary history of recent divergence, CZME and CCYA have more STGD
overlaps compared to other cyanidiophycean species.

line 392

I cannot follow why certain evolutionary constraints result in gene loss. Constraints would function for gene
retention or against gene loss. Is it proper to say “evolutionary pressure” in this case?

[Response]

We thank the reviewer’s comment about the interpretation of gene losses as “evolutionary pressure”. There
are various approaches to interpreting genome evolution (e.g., size, gene content); i) the mutational hazard
hypothesis (genetic drift), ii) the nucleotypic and nucleoskeletal hypotheses, and iii) the genome streamlining
hypothesis (nature selection) (Blommaert, *Proc R Soc B* 2020). In general, genome streamlining (or gene
losses) is associated with avoiding the accumulations of deleterious mutation as a result of large effective

population size (Batut *et al.*, *Nat Rev Microbiol* 2014). Microbial cyanidiophycean species may have a higher
effective population than other eukaryotic species (i.e., macroalgae), resulting in more recombination and
being more affected by selection efficacy. On the other hand, it has been proposed that under stress (i.e.,
nutrient limitation), there is a tradeoff in terms of energy or materials costs allocated to either DNA or other
components (Hessen *et al.*, *Trends Ecol Evol* 2010). According to a genomic survey in the SAR11 clade,
nutrient-limited environments can promote evolutionary pressure, and they were able to reduce the material
costs of replication through genome streamlining (Giovannoni *et al.*, *Science* 2005). In other cases,
approximately 97% of gene deletions exhibited a measurable altered growth phenotype in >1K chemical
genomic assays on the model yeast (Hillenmeyer *et al.*, *Science* 2008; Musso *et al.*, *Genome Res* 2008). Based
on these facts, we expanded this to include the possibility that certain evolutionary constraints would result in
genome reduction or gene loss (environment-dependent conditional dispensability) (Albalat & Cañestro, *Nat*
*Rev Genet* 2016). Because different points of view (for example, neutral theory) make our interpretations
contentious, we toned down and revised the paragraph as follows,

• Page 18, Line 414-424: “Due to the various approaches used to interpret genome evolution (e.g., size,
gene content), gene losses due to "evolutionary pressure" may be a controversial topic; i) the mutational
hazard hypothesis (genetic drift), ii) the nucleotypic and nucleoskeletal hypotheses, and iii) the genome
streamlining hypothesis (nature selection) ⁶⁴. According to genome streamlining hypothesis, studies
have shown that stressful environments (e.g., nutrient-limited) can promote evolutionary pressure to
reduce energy or material costs through genome streamlining both in eukaryotes and prokaryotes ⁶⁵⁻⁶⁸.
Certain evolutionary constraints would result in genome reduction or gene loss (environment-dependent
conditional dispensability) in these cases ⁶⁹. Thus, we propose that the strong evolutionary constraints
imposed by external factors (e.g., heavy metal exposure, thermal stress) resulted in the parallel loss of
functionally equivalent genes.”

-Typos/Wording
line 52

Delete the period from “a. variety of.”

**[Response]**

We corrected the typo as follows,

• Page 3, Line 51: “... inhabit a variety of extreme habitats ...”

line 219

“Linker protein” should be “linker peptide.”

**[Response]**

We corrected the sentence as follows,

• Page 10, line 224: “... linker peptide that connects two ...”

• Page 11, line 226-227 : “... (spacer sequences; linker peptides) was observed among subtelomeric
duplicated proteins, and these linker peptides may ...”

Lines 330 and 354

What does “host” mean in this context?

**[Response]**

We revised the sentence as follows,

• Page 15, Line 338: “... through an ancestral-derived transport ...”

• Page 16, Line 364: “... derived from the eukaryotic ancestor (e.g., arsenic ABC transporter ATPase ...”

Line 381

Spell out “pri-miRNA” as this is the first emergence of this wording.

**[Response]**

We updated the sentence as follows,

• Page 17, Line 398: “... produce primary miRNAs (pri-miRNAs), which ...”

Reviewers' Comments:

Reviewer #1:

Remarks to the Author:

My concerns/comments were addressed. In order for the full potential of this study to be reached, it will be essential that predicted transcript and protein sequences be deposited for the public to include in their own studies, but the authors have ensured that this will be done before publication.

Reviewer #2:

Remarks to the Author:

This is the revised manuscript that now does not contain anecdotal claims. Concerns raised by reviewers are well addressed. I have no more comment to improve this manuscript.

**Point-by-point response**

**Reviewer #1:**

My concerns/comments were addressed. In order for the full potential of this study to be reached, it
will be essential that predicted transcript and protein sequences be deposited for the public to include
in their own studies, but the authors have ensured that this will be done before publication.

**[Response]**

We thank the reviewer for giving insightful comments about our work, as well as for the detailed
recommendations for revising of our manuscript. These suggestions have strengthened the ideas in our
manuscript. In response to the reviewer's comment, we uploaded all the materials used in analyses and
uploaded genome data to an easily accessible public database (NCBI).

**# Reviewer link:**

- • Dryad dataset (doi: 10.5061/dryad.cfxpvnv7b):
<https://datadryad.org/stash/share/wG12T-8kTGCxrTffWD-vTVbRL2IDPvkIUfA2tUQTJus>
- • NCBI (BioProject: PRJNA851236; release data: Aug 31, 2022)
<https://dataview.ncbi.nlm.nih.gov/object/PRJNA851236?reviewer=8a2tavah84stp8hvirm6ts0kq6>

**Reviewer #2:**

This is the revised manuscript that now does not contain anecdotal claims. Concerns raised by
reviewers are well addressed. I have no more comment to improve this manuscript.

**[Response]**

We appreciate the reviewer's interest in our work, as well as the detailed comments and thorough review.
Again, the comments of the reviewer have improved our manuscript.